# Dimerization of GAS2 mediates crosslinking of microtubules and F-actin

Jiancheng An ID[1], Tsuyoshi Imasaki ID[2], Akihiro Narita ID[3], Shinsuke Niwa ID[4], Ryohei Sasaki[1], Tsukasa Makino ID[1], Ryo Nitta[2] & Masahide Kikkawa ID[1✉]

## Abstract

**The spectraplakin family protein GAS2 was originally identified as a growth arrest-specific protein, and recent studies have revealed its involvement in multiple cellular processes. Its dual interaction with actin filaments and microtubules highlights its essential role in cytoskeletal organization, such as cell division, apoptosis, and possibly tumorigenesis. However, the structural basis of cytoskeletal dynamics regulation by GAS2 remains unclear. In this study, we present cryo-electron microscopy structures of the GAS2 type 3 calponin homology domain (CH3) in complex with F-actin at 2.8 Å resolution, thus solving the first type CH3 domain structure bound to F-actin and confirming its actin-binding activity. We also provide the first near-atomic resolution cryo-EM structure of the GAS2-GAR domain bound to microtubules and identify conserved microtubule-binding residues. Our biochemical experiments show that GAS2 promotes microtubule nucleation and polymerization, and that its C-terminal region is essential for dimerization, bundling of both F-actin and microtubules, and microtubule nucleation. As mutations leading to expression of C-terminally truncated GAS2 have been linked to hearing loss, these findings suggest that the disruption of GAS2-dependent cytoskeletal organisation could underlie auditory dysfunction.**

**Keywords** GAS2; Microtubule; Actin; Spectraplakin; Cryo-Electron Microscope
**Subject Categories** Cell Adhesion, Polarity & Cytoskeleton; Structural Biology

## Introduction

The interplay between actin and microtubules (MTs) is crucial for maintaining cytoskeletal structures (Fletcher and Mullins, 2010; Pimm and Henty-Ridilla, 2021; Dogterom and Koenderink, 2019; Rodriguez et al, 2003; Nakos et al, 2022). Spectraplakin primarily mediates the interactions between actin filaments (F-actin) and microtubules, playing key roles in regulating cell polarity, cell-cell junctions, cell division, cell migration, and maintenance of neuron shape (Dogterom and Koenderink, 2019; Huber et al, 2015; Willige et al, 2019). However, the mechanisms and physical principles underlying spectraplakin-mediated actin–microtubule cross-linking are not well understood (López et al, 2014). No structural model currently explains these complex interactions.

Spectraplakins are a family of multi-domain proteins and include Dystonin (BPAG1), MACF1 (ACF7), and growth-arrest-specific 2 (GAS2) family, which comprises GAS2 and GAS2-like proteins 1–3 (Suozzi et al, 2012; Pinto-Costa and Sousa, 2021; Voelzmann et al, 2017). Most spectraplakins contain two N-terminal calponin homology (CH) domains for F-actin binding, plakin and spectrin repeats, and a C-terminal GAR domain for microtubule binding (Fig. EV1A) (Wu et al, 2008; Suozzi et al, 2012). The GAS2 family cross-links F-actin and MTs, which is essential for the regulation of axon morphology, cell apoptosis, and cell division (Willige et al, 2019; Stroud et al, 2014; H. Lee et al, 1999; Zhang et al, 2011). Notably, the GAS2 protein is the most diminutive spectraplakin, comprising a single type 3 CH domain and a GAR domain connected by a hinge linker (H-Linker, Fig. 1A, first panel).

The CH domain, a ~ 110-residues protein module, is known for its roles in actin-binding, signaling, and microtubule binding. The CH domains are highly conserved, with four major helices forming a compact globular structure (Bramham et al, 2002). Three major groups of CH domain have been recognized—CH1, CH2, and CH3—based on sequence alignments and variations in the loop or short helix structures (Keep et al, 1999; Bañuelos et al, 1998). CH1 domains typically have a longer first major helix; the CH2 family features a short helix following the second long major helix, a structure absent in CH1 and CH3; the CH3 domain uniquely contains a short four-residue $3_{10}$ helix after the third main helix (Gimona et al, 2008). CH1 and CH2 domains appear as tandem pairs in MACF1, BPAG1, and Plectin, performing actin binding and bundling functions (Yue et al, 2016; Bañuelos et al, 1998). In contrast, single CH3 domains are found in proteins involved in muscle contraction regulation, such as calponin, and in signal-transduction proteins like Vav and IQGAPs. The interaction of the single CH3 domain with F-actin remains controversial (Castresana and Saraste, 1995; Korenbaum and Rivero, 2002; Bramham et al,

[1]Department of Cell Biology and Anatomy, Graduate School of Medicine, The University of Tokyo, Tokyo, Japan. [2]Division of Structural Medicine and Anatomy, Department of Physiology and Cell Biology, Kobe University Graduate School of Medicine, Kobe, Japan. [3]Structural Biology Research Center, Graduate School of Science, Nagoya University, Nagoya, Aichi, Japan. [4]Graduate School of Life Sciences, Tohoku University, Sendai, Japan. ✉E-mail: mkikkawa@m.u-tokyo.ac.jp

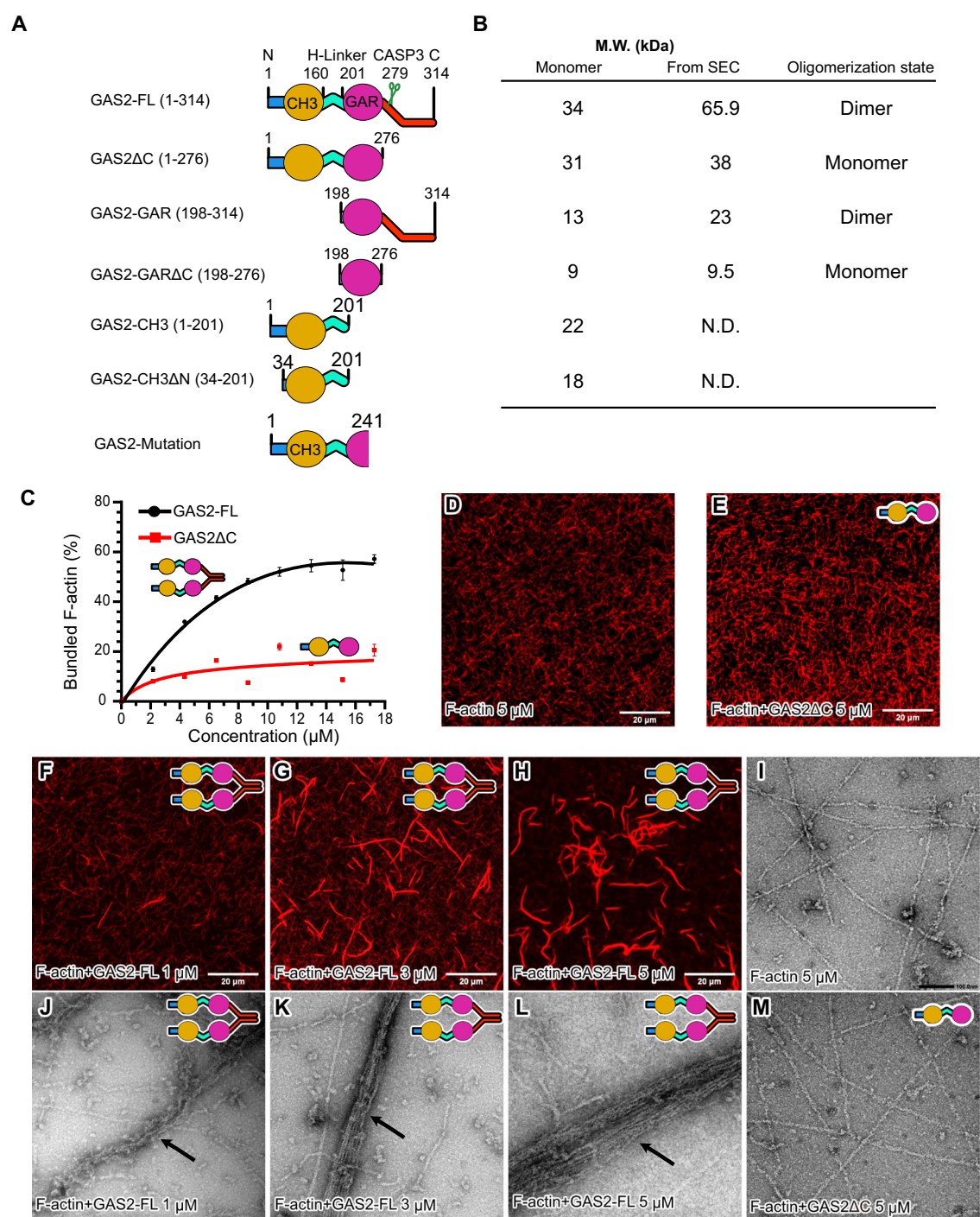

2002; Gimona and Mital, 1998). The differences in F-actin binding among CH1 and CH3 domains are not well understood. Additionally, a special group of single CH domains is present in EB proteins (EB-type CH domain), which specifically bind to the microtubule plus end (Zhang et al, 2015). The mechanisms by which this highly conserved CH domain recognizes the difference between the microtubules and F-actin remain elusive.

The GAR domain consists of approximately 70 amino acids and binds to the MT lattice (Applewhite et al, 2013, 2010; Leung et al, 2002;

Kapur et al, 2012). The GAR domain is highly conserved from *C. elegans* to humans. According to an X-ray crystallographic analysis of the GAR domain of ACF7, it has a unique zinc-binding α/β fold, wherein zinc plays a pivotal role in maintaining the structural integrity of the domain (Lane et al, 2017). The GAR domain is essential for MACF1's microtubule binding and is critical for neuronal development and axon guidance. Mutations in the zinc-binding motif of MACF1's GAR domain disrupt this interaction, resulting in lissencephaly and brainstem hypoplasia (Dobyns et al, 2018).

◀  **Figure 1.   The C-terminal of GAS2 is necessary for dimerization and is essential for F-actin bundling.**

(A) GAS2 constructs used in biochemical assays. The first panel shows that the full-length GAS2 (GAS2-FL) consists of the CH3 and GAR domains, which are connected by a hinge linker (H-Linker). Mouse GAS2-FL is a 314 amino acids protein; the theoretical monomer M.W. (molecular weight) is about 34 kDa. The scissors icon indicates the caspase-3 cleavage site at D279. The second panel shows the GAS2ΔC (residues 1–276), the theoretical monomer M.W. is approximately 31 kDa. The third panel shows the GAS2-GAR domain (residues 198–314); the theoretical monomer M.W. is about 13 kDa. The fourth panel shows the GAS2-GARΔC (residues 198–276); the theoretical monomer M.W. is about 9 kDa. The fifth panel shows the GAS2-CH3 domain (residues 1–201). The sixth panel shows the GAS2-CH3ΔN (residues 34–201). The bottom panel denotes the mutation (c.723+1 G > A) associated with hearing loss, which leads to intron retention and results in a C-terminally truncated GAS2 protein originating from the GAR domain. (B) GAS2 constructs theoretical M.W., computational M.W. of the SEC (Size Exclusion Chromatography) results, and oligomerization state. (C) Dose-dependent bundling curve of F-actin by GAS2-FL or GAS2ΔC for low-speed co-sedimentation assay. F-actin (5 µM as monomer) was incubated with increasing concentrations of GAS2-FL or GAS2ΔC (2–16 µM as monomer), and actin-bundling assays were performed. The amount of bundled F-actin was plotted against the concentration of GAS2-FL or GAS2ΔC for SDS-PAGE results in Fig. EV1H,I. Mean ± SD from three independent experiments is shown. (D–H) Confocal fluorescence microscopy bundle assay in the presence or absence of GAS2-FL or GAS2ΔC. Actin filaments were labeled by Alexa Fluor 568 phalloidin and imaged under confocal microscopy. (I–M) Representative negatively stained electron micrographs of F-actin in the presence or absence of purified GAS2-FL or GAS2ΔC. Black arrows show the F-actin bundle arrays. Source data are available online for this figure.

GAS2, a component of the actin cytoskeleton, is processed by caspase enzymes during apoptosis, which cleaves its carboxyl-terminal (C-terminal) domain. This cleavage leads to significant reorganization of the actin cytoskeleton, altering cell shape and playing a crucial role in the morphological changes associated with cell death (Brancolini et al, 1995). However, the function of the C-terminal is always elusive. Additionally, during the G0 to G1 transition of the cell cycle, GAS2 is rapidly phosphorylated in response to growth factors, resulting in its relocation to membrane ruffles at cell edges and inducing significant microfilament rearrangement (Stroud et al, 2014; Brancolini et al, 1992). In vitro studies further suggest that GAS2 is involved in the differentiation of keratinocytes and muscle cells and can inhibit cell division, resulting in multinucleated cells by microtubule bundling ability (Zhang et al, 2011). In vivo studies demonstrate that GAS2 functions as a death substrate in apoptosis, influencing chondrocyte differentiation and limb myogenesis (H. Lee et al, 1999). Notably, GAS2 is also expressed in Pillar and Deiters' cells, which are crucial for sound wave transduction to hair cells. These cells possess a rigid cytoskeleton of hundreds to thousands of microtubules organized into tightly bundled arrays cross-linked with actin filaments (Fig. EV1B) (Chen et al, 2021). GAS2 colocalizes with these microtubule bundles, and the variants (GAS2 c.723+1G>A; GAS2 c.616−2A>G) introduce a stop codon within the GAR domain (Fig. 1A bottom panel), resulting in disordered microtubule arrays and causing inherited hearing loss (Chen et al, 2021; Zhang et al, 2024). However, its molecular mechanism is poorly understood.

Although GAS2 is essential for cytoskeleton rearrangement during apoptosis, the mechanisms by which GAS2 mediates these changes are not well understood. The functional implications of specific GAS2 mutations causing hearing loss remain unclear due to a lack of structural data and limited functional characterization. To address these gaps, here, we obtained a cryo-EM structure of the F-actin complexed with the CH3 domain of GAS2 at 2.8 Å resolution and a cryo-EM structure of the GAS2-GAR domain decorated on microtubules at 3.2 Å resolution. Our co-sedimentation, negative staining, and TIRF assays demonstrate that GAS2 binds to F-actin, interacts with microtubules, and promotes microtubule nucleation and stabilization. Additionally, GAS2 bundles microtubules and F-actin at low concentrations, suggesting a role in maintaining cell stiffness. Our findings reveal that the C-terminal region of GAS2 is essential for its dimerization, tubulin nucleation, and bundling abilities, providing significant insights into the molecular mechanisms of GAS2-mediated F-actin to microtubule cross-linking. This may pave the way for new therapeutic strategies for diseases involving dysregulated microtubule and actin dynamics.

## Results

### The C-terminal of GAS2 is necessary for dimerization and is essential for F-actin bundling

Full-length GAS2 (GAS2-FL (1–314)) consists of the CH3 and GAR domains interconnected by a hinge linker (H-Linker, Fig. 1A). To evaluate the functional importance of each domain, we generated a series of deletion mutants and assessed them using biochemical assays, TIRF microscopy, and negative stain electron microscopy (EM) (Fig. 1A). Analysis by size exclusion chromatography (SEC) showed that the peak location of GAS2-FL corresponds to 66 kDa, indicating that the protein forms a homodimer (Figs. 1A,B and EV1C–E). On the other hand, GAS2ΔC, lacking the C-terminal domain, appeared as a monomer protein (Figs. 1A,B and EV1C–E), suggesting that the C-terminal domain is important for dimerization. To further identify the minimum region required for dimerization, we generated GAS2-GAR domain constructs with and without the C-terminal region (GAS2-GARΔC) to further identify the minimum region for dimerization and performed SEC experiments (Fig. 1A). The results showed that GAS2-GAR is a dimer, while GAS2-GARΔC is a monomer (Figs. 1B and EV1C–E). These findings demonstrate the critical role of GAS2's C-terminal domain in its dimerization.

We utilized a co-pelleting assay to investigate the interaction between full-length GAS2 and actin filaments in vitro. Both full-length GAS2 and its CH3 domain (Fig. 1A, fifth panel, amino acids 1–201) were co-sedimented with actin filaments (Fig. EV1F,G), indicating that the CH3 domain is responsible for binding to F-actin. We performed a low-speed co-sedimentation assay to test whether GAS2 promotes F-actin bundling (Bähler and Greengard, 1987). F-actin alone does not sediment at low centrifugal forces ($9000 \times g$) (Fig. EV1H, leftmost lane). However, increasing concentrations of full-length GAS2 resulted in a concentration-dependent increase in the amount of actin in the pellet, indicating GAS2-induced F-actin bundling (Figs. 1C and EV1H).

In contrast, when GAS2ΔC was incubated with F-actin, the amount of actin in the pellet did not increase to the same extent as

**Table 1. Data collection, refinement, and validation statistics.**

| Data collection and processing | | | |
|---|---|---|---|
| Complex structure | GAS2-CH3-F-actin | GAS2-GAR-MT | GAS2-FL-MT |
| Magnification | 105,000 | 105,000 | 105,000 |
| Voltage (kV) | 300 | 300 | 300 |
| Electron exposure (e-/Å²) | 48.93 | 48.79 | 49.92 |
| Defocus range (μm) | −2.4 to −1.2 | −1.8 to −1 | −1.8 to −1 |
| Pixel size (Å) | 0.83 | 0.83 | 0.83 |
| Number of images taken | 6378 | 3322 | 261 |
| Symmetry imposed | Pseudo-Helical | Pseudo-Helical | Pseudo-Helical |
| Initial particles number | 632,290 | 70012 | 22,841 |
| Final particles number | 412,176 | 44,922 | 5616 |
| Map resolution (Å) | 2.8 Å | 3.2 Å | 4.39 Å |
| FSC threshold | 0.143 | 0.143 | 0.143 |
| Map sharpening B factor (Å²) | −111 | −89 | |
| **Refinement** | | | |
| Initial model used | AlphaFold: AF-P11862-F1 (GAS2-CH3) | PDB: 1V5R (GAS2-GAR) PDB: 3JAT (tubulin) | |
| Model resolution (A˚)/FSC threshold | 2.88/0.5 | 3.84/0.5 | |
| **Model composition** | | | |
| Non Hydrogen atoms | 13,802 | 7434 | |
| Protein residues | 1,756 | 937 | |
| Ligands | MG:4/ADP:4 | GTP:2/ZN:1/MG:2 | |
| **R.m.s. deviations** | | | |
| Bond lengths (A˚) | 0.006 | 0.005 | |
| Bond angles (˚) | 0.689 | 0.806 | |
| **Validation** | | | |
| MolProbity score | 1.40 | 2.06 | |
| Clash score | 4.92 | 16.40 | |
| **Ramachandran plot** | | | |
| Favored (%) | 98.16 | 95.06 | |
| Allowed (%) | 1.84 | 4.94 | |
| Outliers (%) | 0.00 | 0.00 | |

observed with full-length GAS2 (Figs. 1C and EV1I). At 16 μM concentration, full-length GAS2 bundled 60 ± 1.9% of F-actin, while GAS2ΔC only bundled 23 ± 2.6% under the same conditions (Fig. 1C). This suggests that the C-terminal domain of GAS2 is necessary for efficient F-actin bundling. Fluorescence microscopy further supported these findings; GAS2 addition led to longer, thicker filaments with enhanced fluorescence, indicating actin bundling (Fig. 1F–H). Conversely, the GAS2ΔC failed to induce

significant changes (Fig. 1E), similar to the bare F-actin control (Fig. 1D). Negative stain electron microscopy further confirmed these observations; in the absence of GAS2-FL, actin filaments remained disordered (Fig. 1I), whereas in its presence, actin filaments bundled into higher-order structures (Fig. 1J–L). GAS2ΔC failed to induce bundling (Fig. 1M).

In conclusion, our results prove that the CH3 domain of GAS2-FL can interact with actin filaments. The low-speed co-sedimentation assay demonstrated that GAS2-FL can bundle actin filament and that the C-terminal (residues 277–314) of GAS2 is essential for this activity.

## Cryo-EM structure of GAS2 CH domain bound to F-actin

To understand how GAS2 recognizes F-actin through its CH3 domain at the molecular level, we utilized cryo-EM single-particle analysis. We used GAS2-CH3 (residues 1–201, Fig. 1A fifth panel), the minimum construct that can bind to F-actin, and obtained a high-quality density map with a local resolution gradient better than 3.0 Å, as judged by the FSC curve (Table 1, Figs. 2A and EV2A). To model the atomic structure, we used the AlphaFold2 prediction structure of GAS2-CH3 as a starting model (AF2 model No. AF-P11862-F1) (Jumper et al, 2021). The AF2 model fits very well, except for a few flexible loops. After several rounds of manual modeling and refinement, we obtained the final atomic model (Figs. 2C and EV2B). The high-resolution areas of the GAS2-CH3 domain, located at the interface with F-actin, allowed accurate modeling and elucidation of their interactions (Fig. EV2B,C). Conversely, the lower-resolution regions corresponding to the loops of GAS2-CH3 were situated distal to the F-actin interface and exhibited poor map fitting, suggesting their intrinsic flexibility (Fig. EV2D).

Our atomic model identified three main actin-binding sites in the GAS2-CH3 domain, namely ABD1 (amino acids 13–31), ABD2' (amino acids 107–111), and ABD2 (amino acids 127–145) (Figs. 2D and EV2B). At ABD1 (amino acids 13–31), the N-terminal loops or helices of the GAS2-CH3 domain mainly interact with actin (n) by hydrophobic or electrostatic environment (Figs. 2B,D,I,J and EV2B). At ABD2', the n + 2 position of actin Y143, I345, L349, and n position of actin M44 at the D-loop form a hydrophobic pocket that enfolds F108 and F109 of the GAS2-CH3 domain (Figs. 2B,D,E and EV2B). ABD2 had electrostatic interactions with the n position of actin between actin H87 and GAS2 T129, actin K61 and GAS2 E133, actin E93 and GAS2 R144 (Figs. 2B,D,F–H and EV2B).

To validate our structural model, we generated point mutation proteins and analyzed them using an F-actin co-pelleting assay. We introduced mutations L16D, Y22D, W25E, and GAS2-CH3ΔN (Fig. 1A sixth panel) targeting to disrupt ABD1 hydrophobic interactions, one mutation (F108D) targeting the hydrophobic pocket in ABD2', and two mutations (E133R and R144E) targeting electrostatic interaction in ABD2 (Figs. 2K and EV2E,F). Compared to the wild-type CH3 domain, Y22D, GAS2-CH3ΔN, F108D, E133R, and R144E showed significantly lower affinity for F-actin (Figs. 2K and EV2E,F). Y22 (ABD1) and F108 (ABD2') are critical residues that directly interact with the hydrophobic pocket of F-actin. Introducing charged mutations, such as Y22D and F108D, significantly compromises these hydrophobic interactions. Furthermore, charge-reversing mutations, including E133R and R144E, profoundly perturb the original electrostatic interactions. These

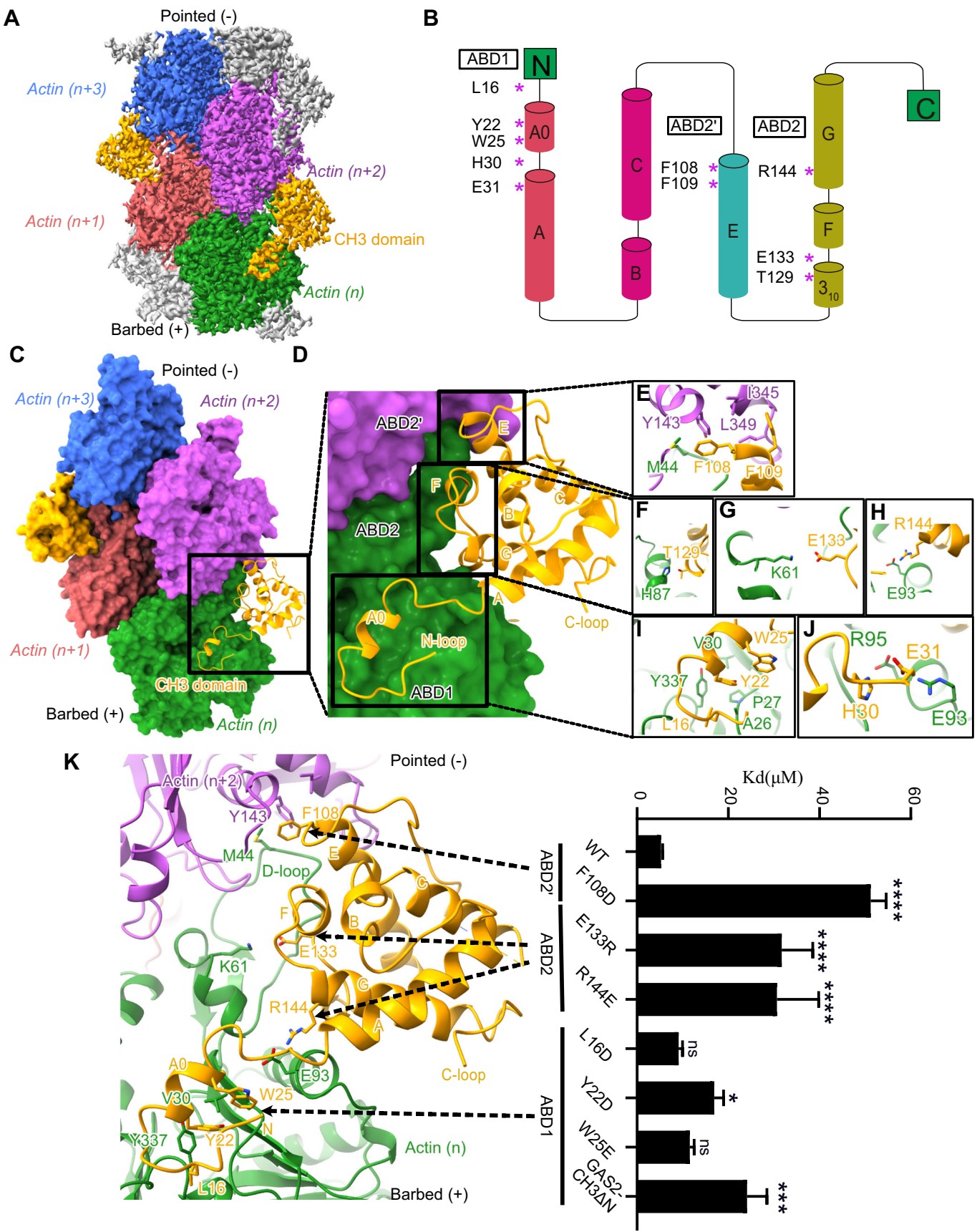

**Figure 2.   Cryo-EM map and model of GAS2-CH3 domain bound to F-actin.**

(A) Reconstruction cryo-EM map of GAS2-CH3 domain decorated on F-actin at a 2.8 Å resolution. 4 Actin binding subunits (marked as n series from bottom to top) are presented in green, Indian red, medium orchid, royal blue, and GAS2-CH3 domain is presented in orange, respectively. (B) Topology structure of GAS2-CH3 domain and the main interaction amino acids in the topology structure. Four main helices are labeled with A, C, E, and G; short helices including A0, B, F and a short $3_{10}$ helix. (C) The atomic model of the GAS2-CH3 domain is decorated on F-actin with the same color scheme as in (A). The GAS2-CH3 domain in orange interacts with two adjacent actin monomers, thus following the actin helical pattern. (D) Closer view of the GAS2-CH3-F-actin model, there are mainly three actin-binding sites for the GAS2-CH3 domain, respectively ABD1, ABD2′, and ABD2, which are shown in the black box. (E–J) Residual level information of key amino acids interacting with actin monomers from ABD1: L16, Y22, and W25 in (I), H30 and E31 in (J); ABD2′: F108, F109 in (E); ABD2: T129 in (F), E133 in (G), and R144 in (H). (K) The histogram of $K_d$ values for GAS2-CH3 WT and mutants of representative Coomassie-stained actin co-sedimentation assay. $K_d$ values are shown in (Fig. EV2E), and the fitting curves are shown in (Fig. EV2F). Raw SDS-PAGE gels are shown in source data. Bars = mean ± SD. from three independent experiments, *(p-value < 0.05), ***(p-value < 0.001), ****(p-value < 0.0001) significantly different from WT in a one-way ANOVA with Dunnett's multiple comparisons test, ns, not significant. p-value from top to bottom: 1.4E−9, 6.1E−6, 1.1E−5, 0.8359, 0.0258, 0.3997, 0.0004. Source data are available online for this figure.

results collectively highlight that the core binding affinity of GAS2-CH3 to F-actin is predominantly mediated by the ABD1, ABD2′, and ABD2 sites, underscoring the functional importance of these residues in actin binding. Additionally, the results confirm that a single GAS2-CH3 domain is sufficient for interaction with F-actin.

## GAS2 bundles microtubule by binding to MT through GAR domain

Utilizing co-pelleting assays, we confirmed that both full-length GAS2 and its GAR domain (residues 198–314) exhibit binding affinity to taxol-stabilized MTs (Fig. EV3A,B). Dark-field microscopy directly visualized MT bundling by different GAS2 constructs. We found that GAS2-FL and GAS2-GAR, particularly with an intact C-terminal region, induced dense clustering and bundling of MTs (Movie EV1 for the bare MT; Movie EV2 for GAS2-FL; Movie EV3 for GAS2-GAR). In contrast, GAS2-ΔC and GAS2-GARΔC without the C-terminal failed to produce the effects (Movie EV4 for GAS2ΔC; Movie EV5 for GAS2-GARΔC). Electron microscopy further elucidated well-ordered MT bundles formed in the presence of GAS2-FL compared with bare MT (Fig. 3A,D–F), emphasizing the crucial role of the C-terminal region in MT bundling. Additionally, the GAS2-GAR domain demonstrated significant MT bundling ability (Fig. 3G–I). In contrast, GAS2ΔC and GAS2-GARΔC, which lack the C-terminal region, exhibited an inability to induce MT bundling (Fig. 3B,C). These findings underscore the essential role of the C-terminal region in GAS2-mediated MT bundling.

We measured the diameter of MTs and MT bundles in micrographs and found that the average diameter of bare MTs was about 30 nm. However, in the presence of GAS2-FL or GAS2GAR, the average width of MT bundles widened significantly to approximately 150 nm or 250 nm, respectively (Fig. 3J). Conversely, there was minimal change in MT diameter with GAS2ΔC or GAS2-GARΔC presence when these proteins lost the C-terminal (Fig. 3J). In summary, our results furnish compelling evidence that the C-terminal region of GAS2 is pivotal for bundling MT.

## Cryo-EM structure of GAS2-GAR-microtubule complex

To further investigate the interaction between GAS2 and MT, we used the GAS2-GAR domain construct to elucidate the GAR structure bound to microtubules by single-particle cryo-EM analysis. After eliminating poorly decorated particles, 2D class averages exhibited that the GAS2-GAR domain distributes along microtubule surfaces with 8 nm intervals (Fig. EV3C,D). We

employed MiRP ver2 to analyze the microtubule, which exhibits a pseudo-symmetric nature with a seam (Cook et al, 2020). Our analysis achieved a 3.2 Å resolution of 14-pf microtubules complexed with the GAS2-GAR domain (Fig. EV3C). Local resolution estimates varied from 2.5 to 3.5 Å for tubulins and 3.5 to 4.5 Å for GAR domain molecules (Fig. EV3E). The density map reveals that the GAR domain is inserted into the tubulin intradimer groove (Fig. 4A,B). Notably, the resolutions of N- and C-termini and certain loops of the GAS2-GAR domain dip below 4 Å, possibly due to its flexibility (Fig. EV3E).

For a higher-resolution analysis of the GAS2-GAR-MT complex, we reprocessed the cryo-EM images using symmetry expansion and achieved a 2.8 Å overall resolution. This improved map enabled us to build an atomic model and observe amino acid sidechains at the GAR-MT interface (Appendix Fig. S1A). However, the distinction between α- and β-tubulin became less pronounced (Appendix Fig. S1B).

To elucidate the interaction between GAR and MT, we used the previously reported NMR structure (PDB: 1V5R, Appendix Fig. S1C) and αβ-tubulin heterodimers (PDB: 3JAT) as starting models. α- and β-tubulin are similar, but we distinguished them based on the difference in the length of the S9-S10 loop (Fig. EV3F). Manual modeling and iterative refinement resulted in the final atomic model of the GAS2-GAR-MT complex (Figs. 4C and EV3G; Appendix Fig. S1D). The MT-binding surface of GAR was modeled well to fit into the density map, while the flexible N- and C-termini and some loops were outside the map, such as loop E225 to Y232 (Appendix Fig. S1E). The atomic model of GAS2-GAR bound to tubulin reveals that the GAR domain interacts with microtubules primarily through the β3-β4-β5-α2 regions and associated loops. Notably, β3 of the GAR domain forms hydrophilic interactions with β-tubulin H5 and α-tubulin H11′. Specifically, GAS2 R242 potentially engages in a salt bridge with α-tubulin E411 (Fig. 4E). Additionally, GAS2 R242 faces β-tubulin D163, possibly facilitating interaction (Fig. 4E). Similarly, GAS2 β4 R252's proximity to β-tubulin D199 implies a potential salt bridge consistent with the observed density (Fig. 4F).

Within the β4-β5 loop of the GAS2-GAR domain, there are three glycine residues (G254, G255, G256), which are highly conserved across GAR domain homologs (Appendix Fig. S2A). The β4-β5 loop forms a hairpin structure (G-hairpin) and penetrates into the groove of α-β tubulin intradimer (Fig. 4F). Furthermore, α-tubulin H11′-W407, Y408, V409, and G410 form a hydrophobic pocket, into which GAR domain β5 W257 snugly fits into this pocket alongside α-tubulin W407 (Fig. 4G). The density map and atomic models reveal GAR domain α-helix2 K266 closes to

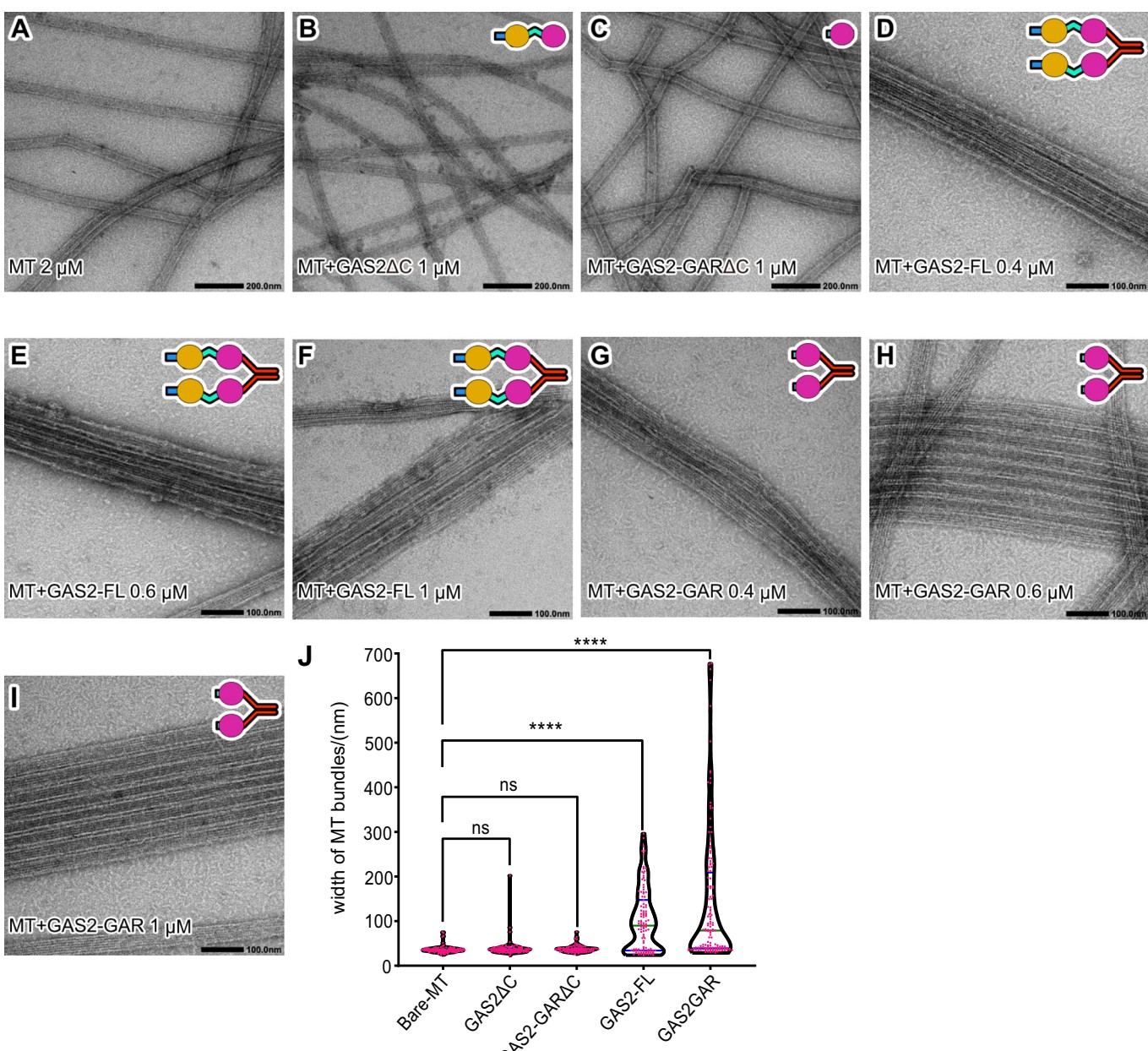

**Figure 3.  Negative staining electron microscopy reveals the GAS2 MTs bundling ability.**

(**A**) Bare MT is disordered as individual filaments from negative staining micrographs. (**B**) GAS2ΔC did not bundle MT, and most MTs are disordered individual filaments from negative staining electron micrographs. (**C**) GAS2-GARΔC did not bundle MT, and most of MTs are disordered individual from negative staining electron micrographs. (**D–F**) GAS2-FL gathered MT to well-ordered bundles at different concentrations from negative staining electron micrographs. (**G–I**) GAS2-GAR domain gathered MT to well-ordered bundles from negative staining micrographs at different concentrations. (**J**) The width of MTs or MTs bundles was measured by ImageJ. In one micrograph, all MTs or MTs bundles width were measured. For control, bare-MT $N = 114$ MTs or MTs bundles were checked; For GAS2ΔC $N = 111$ MTs or MTs bundles were checked; For GAS2-GARΔC $N = 103$ MTs or MTs bundles were checked; For GAS2-FL $N = 108$ MTs or MTs bundles were checked; For GAS2-GAR $N = 137$ MTs or MTs bundles were checked. One-way ANOVA with Dunnett's multiple comparisons test has been used. ****$p$-value < 0.0001, ns: no significant difference, $p$-value from left to right: 0.999, 0.999, 3.8E−9, 1.0E−15. All the detailed values are shown in Source data. Source data are available online for this figure.

β-tubulin E431 and H267 proximity to β-tubulin E196, suggesting electrostatic interactions (Fig. 4H). Additionally, GAR domain β3 K237 is poised to interact with β-tubulin E196 (Fig. 4H). From the structure analysis, we concluded that the GAR domains interact with microtubules through β3-β4-β5-α2 regions.

To validate the GAS2-GAR-MT interactions, we generated a series of point mutants to disrupt observed interactions potentially important for binding. We performed an in vitro co-precipitation assay with taxol-stabilized microtubules. As wild-type GAS2-GAR exhibited $K_d$ of ~0.69 μM, the GAS2-GAR R242E mutation reduced

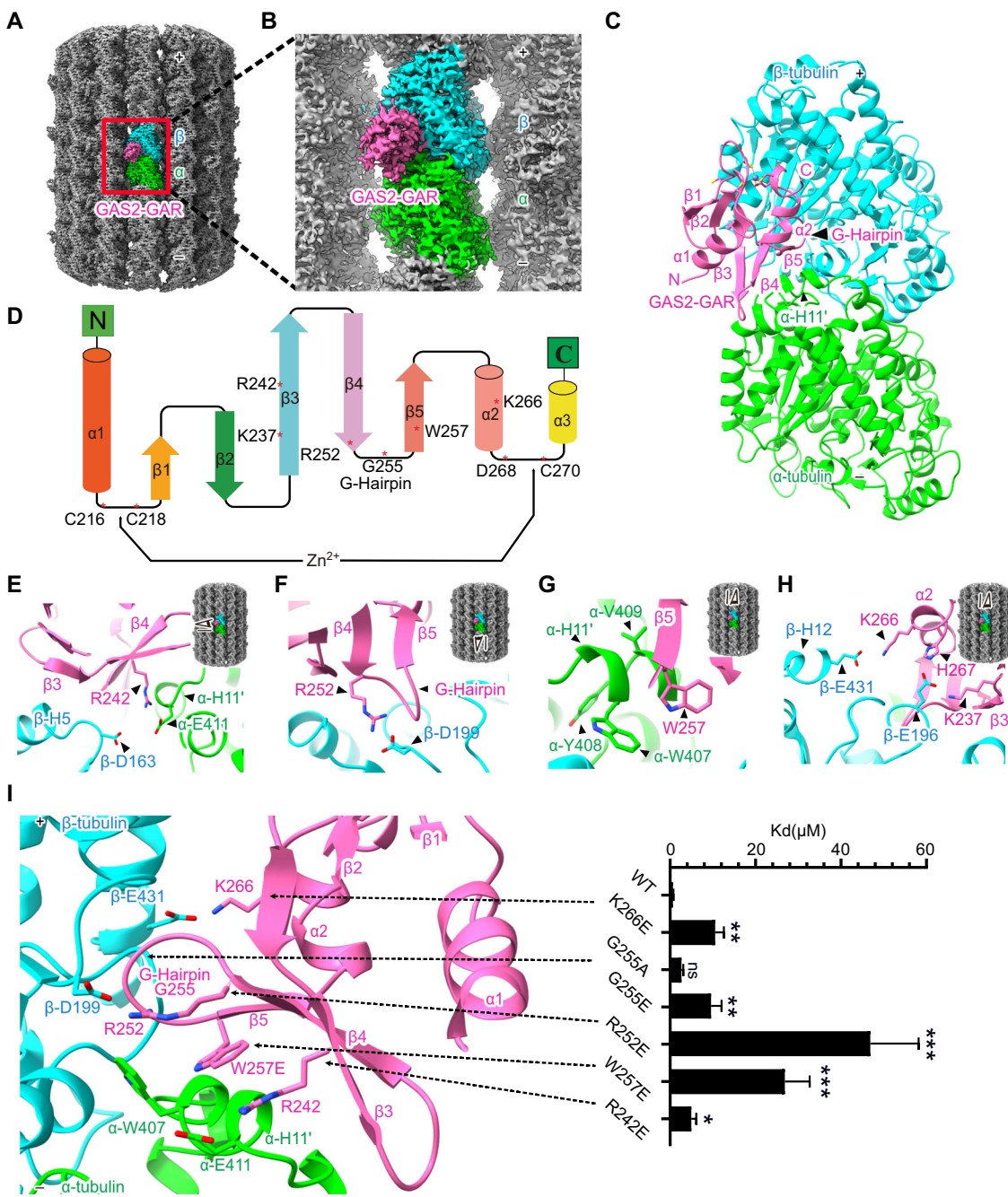

**Figure 4. The 3.2-Å resolution cryo-EM structure of the GAS2-GAR domain bound to the microtubule.**

(A) Cryo-EM map of 14-protofilament GMPCPP-stabilized microtubule decorated with GAS2-GAR domain. α-tubulin: lime, β-tubulin: cyan, and GAS2-GAR domain: hot-pink. (B) Enlarged view of the Cryo-EM map of the red box in (A). (C) Atomic model of GAS2-GAR-tubulin complex. The G-hairpin stench into the intradimer groove of the tubulin. The zinc-binding motif of the GAS2-GAR domain is shown in the enlarged view. (D) Topology structure of GAS2-GAR domain and the main interaction amino acids in secondary structure. All these interactions are located in β3-β4-β5-α2 regions and the loops between β3-β4-β5-α2. (E) The long β3 of the GAS2GAR domain spans from β-tubulin to α-tubulin. The R242 of the GAR domain is close to H11' of α-tubulin's E411 and β-tubulin D163. (F) The G-hairpin penetrates into α-β tubulin intradimer. The R252 is close to β-tubulin D199. (G) α-tubulin H11' W407, Y408, V409, and G410 form a hydrophobic pocket. W257 in β5 of the GAR domain nestles into α-tubulin H11' hydrophobic pocket and is close to α-tubulin W407. (H) K266 closes to β-tubulin E431, K237 in β3 and H267 in α-helix 2 of the GAS2-GAR domain is close to E196 in β-tubulin. (I) The histogram of $K_d$ values for GAS2-GAR WT and mutants of representative Coomassie-stained co-sedimentation assay. $K_d$ values are shown in (Fig. EV3H), and the fitting curves are shown in (Fig. EV3I). Raw SDS-PAGE gels are shown in source data. Bars = mean ± SD. from three independent experiments, *(p-value < 0.05), **(p-value < 0.01), ***(p-value < 0.001) significantly different from WT in an unpaired with two-tailed t test, ns: no significant difference, p-value from top to bottom: 0.0058, 0.1252, 0.0039, 0.0007, 0.0187. Source data are available online for this figure.

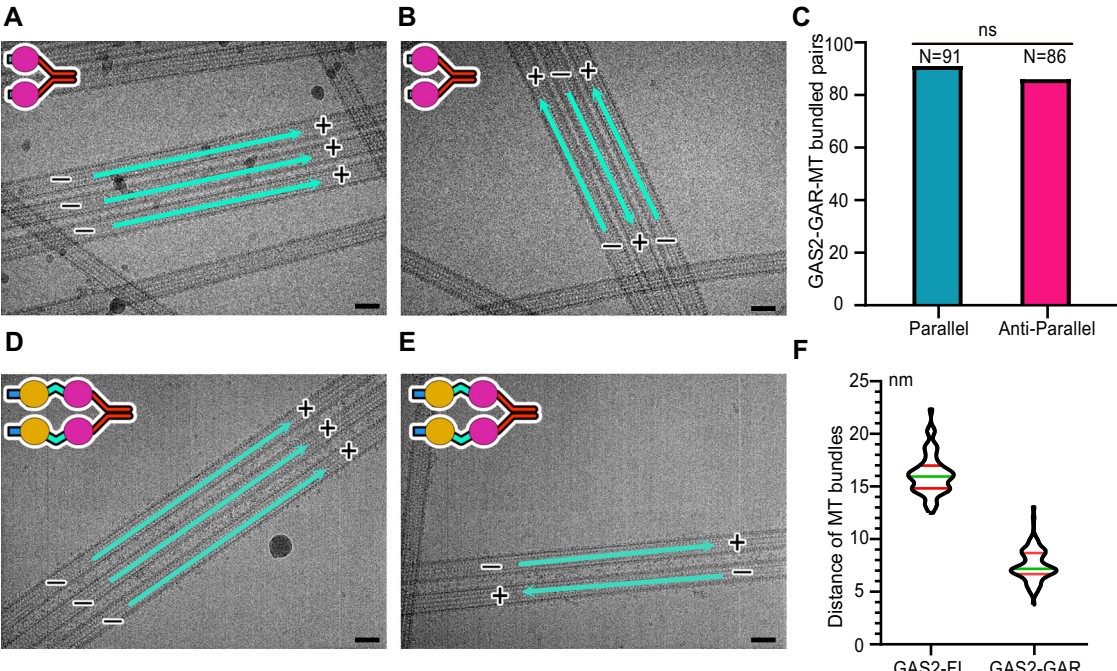

**Figure 5. GAS2 controls the inter-bundle distance of the MT.**

(A, B) Microtubule polarity was traced in the GAS2-GAR-MT cryo-EM micrographs. Representative EM images show the parallel bundled MTs in (A) and the antiparallel bundled MTs in (B). Cyan arrows represent the polarity of MT from minus to plus. (C) Histogram of GAS2-GAR-MT bundled pairs. In total, $N = 177$ MT bundling pairs were checked, $N = 91$ MT pairs were bundled in a parallel way, and $N = 86$ MT pairs were bundled in an antiparallel way. Unpaired with two-tailed t test, ns: no significant difference, $p$-value = 0.8613. (D, E) Microtubule polarity was traced in the GAS2-FL-MT cryo-EM micrographs. Representative EM images show the parallel bundled MTs in (D) and the antiparallel bundled MTs in (E). Cyan arrows represent the polarity of MT from minus to plus. (F) The distance between the bundled MTs was measured in cryo-EM micrographs. $N = 63$ MT bundled pairs were checked in GAS2-FL-MT cryo-EM micrographs (Minimum = 12.5 nm, Maximum = 22.37 nm), $N = 89$ MT bundled pairs were checked in GAS2-GAR-MT cryo-EM micrographs (Minimum = 3.785 nm, Maximum = 13.07 nm). Scale bar = 20 nm. All the detailed values are shown in Source data. Source data are available online for this figure.

binding activity to ~5 μM (Figs. 4I and EV3H,I). GAS2-GAR R252E mutation significantly impacted the binding ability to $K_d$ of ~47 μM (Figs. 4I and EV3H,I), suggesting the importance of the interaction between GAS2-GAR R252 and β-tubulin D199. Additionally, mutation W257E resulted in $K_d \sim 27$ μM, suggesting disruption of hydrophobic interactions against α-tubulin W407. Mutating glycine residues at G-hairpin to alanine (G255A) or glutamic acid (G255E) decreased binding ability significantly ($K_d \sim 2.69$ μM for G255A, ~9.60 μM for G255E, Figs. 4I and EV3H,I). GAS2-GAR K266E at α-helix 2 mutation disrupting interfaces with β-tubulin E431 also reduced the binding activity $K_d$ of ~10.6 μM (Figs. 4I and EV3H,I). From these analyses, we concluded that β3-β4-β5-α2 regions are vital for the GAS2-GAR domain to bind to MT.

## GAS2 controls the inter-bundle distance of the MT

In our GAS2-GAR-MT structure, the C-terminal density remained elusive due to its inherent flexibility. We also successfully resolved the full-length GAS2-FL-MT structure at 4.39 Å; however, the C-terminal and the hinge linker between the CH3 domain of the GAS2 protein remained elusive (Appendix Fig. S2B). This suggests that the flexibility of the hinge linker and C-terminal allows for the optimal orientation of the CH3 and GAR domains when the GAS2 protein mediates the cross-linking of two filaments.

To investigate how GAS2 organizes MT bundles, we compared the polarity of the bundled MT by out scripts (Fig. 5A,B represent the GAS2-GAR domain bundle MT polarity, arrows represent the polarity of MT from minus to plus, Fig. 5D,E represent GAS2-FL). The ratio of parallel and antiparallel bundle MT pairs has no significant difference (Fig. 5C), which illustrates the ability of the GAS2 protein to bundle MT in either orientation. The bundling MT-to-MT distances observed in the Pillar and Deiters' cells are about 120–150 Å (Fig. EV1B) (Saito and Hama, 1982; Tolomeo and Holley, 1997). From our cryo-EM micrographs, the distance between two bundled MTs ranges from 40 to 130 Å in the presence of the GAS2-GAR domain, while it ranges from 120 to 220 Å in the presence of GAS2-FL protein (Fig. 5F). These results indicate that this MT-to-MT distance is consistent with Pillar or Deiters' cells bundling MT and indicate the dimerization GAS2 can adjust the inter-filament distances by the flexible loops.

## GAS2-GAR nucleates and stabilizes the MT

We investigated the time-dependent MT formation using negative stain electron microscopy (EM). Initially, 3 μM tubulin was incubated with 3 μM GAS2-GAR for 10 min on ice, followed by incubation at 37 °C for 1 min. This led to numerous ring formations, whose diameters are ~50 nm (Fig. 6A). Over time, we observed a significant increase in

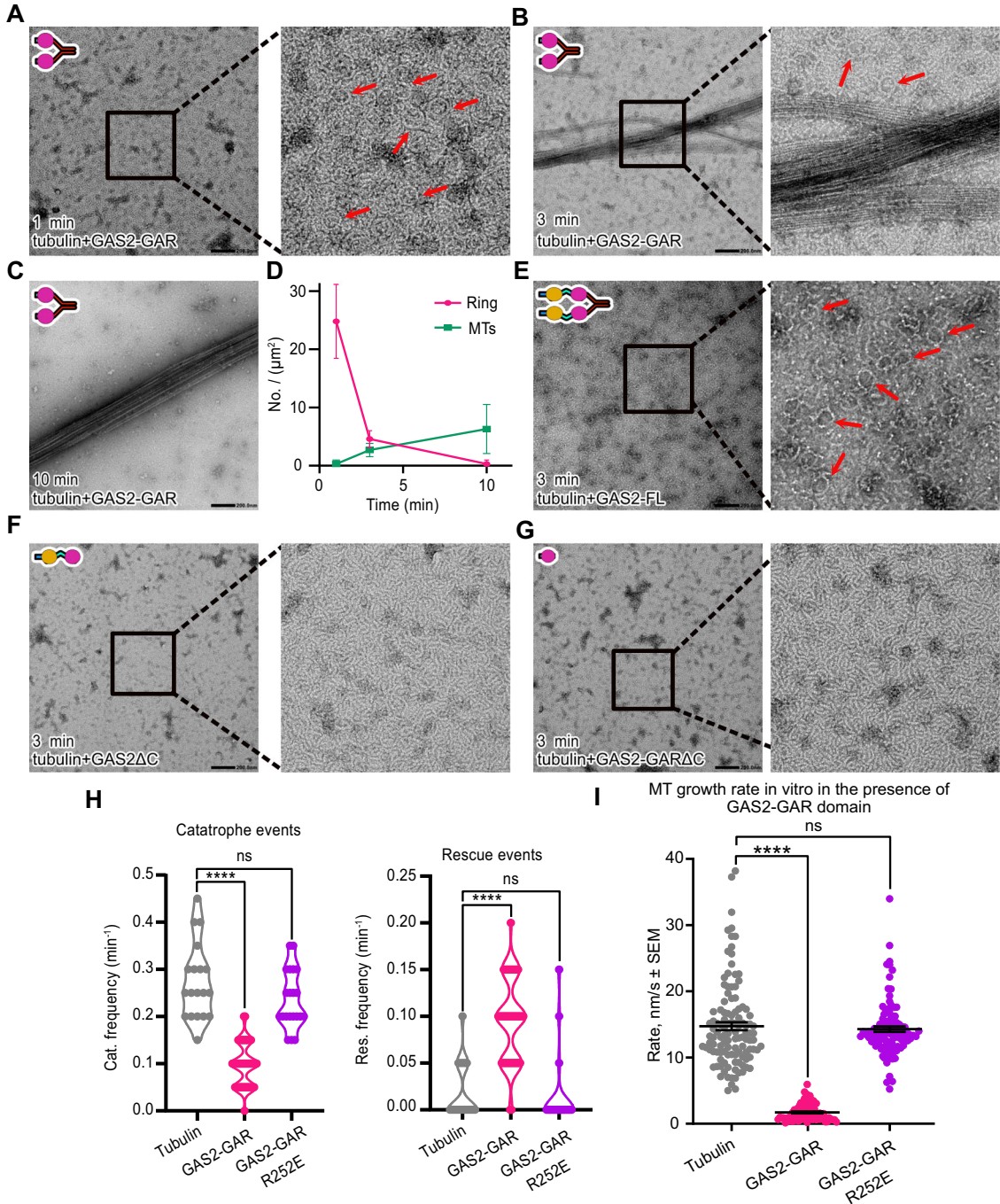

**Figure 6. GAS2-GAR nucleates and stabilizes the MT.**

(A–C) Representative negative stain EM micrographs of 3 µM tubulin polymerization with 3 µM GAS2-GAR at different time points. Red arrows indicate the tubulin ring structures. (D) Plots of the number of tubulin rings (pink) and microtubules (green) at different time points (mean ± SD, from 10 independent views). All the detailed values are shown in Source data. (E) Representative negative stain EM micrographs of 3 µM tubulin polymerization with 3 µM GAS2-FL at 3 min. Red arrows indicate the tubulin ring structures. (F) Representative negative stain EM micrographs of 3 µM tubulin polymerization with 3 µM GAS2ΔC at 3 min. (G) Representative negative stain EM micrographs of 3 µM tubulin polymerization with 3 µM GAS2-GARΔC at 3 min. (H) Effects of GAS2-GAR on the frequency of catastrophes and rescues in microtubules (All the detailed values are shown in Source data). Microtubules were assembled from 10 µM tubulin in the presence of 100 nM GAS2-GAR-GFP or GAS2-GAR-R252E-GFP. $p$-value from left to right for catastrophe events: 1.0E−15, 0.071; $p$-value from left to right for rescue events: 9.8E−10, 0.9206. (I) Microtubule growth rate in vitro in the presence or absence of GAS2-GAR-GFP or GAS2-GAR-R252E-GFP. Black bars show mean ± SEM. $p$-value from left to right: 1.0E−15, 0.728. In (H) and (I), for control group bare-MT $N = 19$ MTs were checked. For GAS2-GAR-GFP, $N = 50$ MTs were checked. For GAS2-GAR-R252E-GFP, $N = 21$ MTs were checked. One-way ANOVA with Dunnett's multiple comparisons test has been used. ****$p < 0.0001$. $p$ values were calculated relative to the tubulin-alone condition, ns: no significant difference. All the detailed values are shown in Source data. Source data are available online for this figure.

microtubules and a reduction in ring structures (Fig. 6B–D). After 10 min at 37 °C, the rings disappeared entirely, indicating their utilization in microtubule growth processes (Fig. 6A–D). In contrast, no ring structures were detected in the tubulin-alone control, even after incubation at 37 °C for 10 min (Fig. EV4A,B). Moreover, microtubules were not observed in the tubulin-alone control until 30 min of incubation, and even then, only a small number of microtubules were detected (Fig. EV4C). These results demonstrate that the dimerization of GAS2-GAR induces longitudinal tubulin oligomerization, facilitating microtubule nucleation and polymerization.

Next, to investigate which domains are important, we incubated 3 μM tubulin with 3 μM GAS2-FL, GAS2ΔC, or GAS2-GARΔC for 3 min. We observed the same ring structures in the presence of GAS2-FL from the negative stain EM micrographs (Fig. 6E). In contrast, no ring structures or microtubules were observed in the presence of GAS2ΔC or GAS2-GARΔC (Fig. 6F,G). These findings suggest that the dimerization of GAS2 constructs has a more prominent effect on tubulin polymerization.

To investigate how GAS2-GAR affects microtubule dynamics, we conducted total internal reflection fluorescence (TIRF)-based assays with microtubules nucleated from randomly oriented short seeds in the presence of pure tubulin (10 μM) and GAS2-GAR (100 nM). We analyzed the frequency of catastrophe (transition from growth to shrinkage) and rescue (transition from shrinkage to growth) events (Fig. EV4D). Compared to tubulin-alone (Movie EV6), GAS2-GAR significantly inhibited catastrophes and promoted rescue events (Figs. 6H and EV4E,F, Movie EV7). GAS2-GAR also inhibited the MT growth rate, because microtubules grew at a rate of 14.73 ± 0.60 nm/s in the absence of GAS2-GAR, while it is 1.71 ± 0.15 nm/s in the presence of 100 nM GAS2-GAR (Fig. 6I). Interestingly, in the presence of GAS2-GAR, the MTs tend to pause after polymerization, and the average pause time is significantly longer, further illustrating that GAS2-GAR promotes microtubule stabilization (Fig. EV4H). We further validated our findings by examining the inhibitory effects of the GAS2-GAR domain in comparison with the GAS2-GAR-R252E mutant, which is unable to bind microtubules (Fig. EV4G, Movie EV8). The rescue and catastrophe frequencies remained unaffected in the presence of the GAS2-GAR-R252E mutant (Fig. 6H). Additionally, the GAS2-GAR-R252E mutation exhibited no significant impact on the polymerization rate (14.29 ± 0.40 nm/s) and pausing behavior of MT (Figs. 6I and EV4H).

### GAS2 cross-links F-actin and microtubule

Our structural and biochemical analyses unveil GAS2's ability to interact with F-actin via the CH3 domain, facilitating filament bundling, and with MT via the GAR domain, promoting MT bundling. This suggests GAS2's potential as a crosslinker between F-actin and MT. To test this idea, we mixed GAS2, F-actin, and MT, incubated them at 25 °C for 50 min, and performed negative stain EM. In the absence of GAS2-FL, there are individual MT and F-actin filaments, and they are not connected or bundled (Figs. 1I, 3A and 7A). On the other hand, the presence of GAS2-FL led to the formation of F-actin-MT bundles, with large F-actin bundles surrounding the MT (Fig. 7B,D). In the presence of GAS2ΔC, the single F-actin filaments were also tethered to the MT; however, we did not see the F-actin bundling cluster surrounding the MT (Fig. 7C,D).

Next, we used low-speed pelleting assays, which can capture only cross-linked actin and MT, to further investigate the cross-linking effects of GAS2 protein on F-actin and MT. After incubating MT and/or F-actin with or without GAS2-FL protein, the mixtures were centrifuged for 10 min at 800 × g, diagnosed by SDS-PAGE, and quantified. Small amounts of pelleted actin were observed in F-actin only, MT and F-actin, and F-actin and GAS2-FL reactions (Fig. 7E columns 1, 2, and 3, Fig. EV4I). Similarly, small amounts of tubulin pellet were observed in MT alone, MT and F-actin, and MT and GAS2-FL reactions (Fig. 7E columns 2, 5, and 6, Fig. EV4J). However, in the presence of GAS2-FL, MT, and F-actin, substantial amounts of both actin and tubulin were observed in the pellet (Fig. 7E column 4, Fig. EV4I–K), suggesting that GAS2 cross-linked actin and MT. These results are consistent with the observation of the GAS2-F-actin-MT complexes by negative staining microscopy.

## Discussion

### GAS2 interacts with F-actin

GAS2, which contains only one CH3 domain, belongs to the 1×CH protein family. Our biochemical analysis revealed that a single CH domain is sufficient for actin binding. Our high-resolution cryo-EM structure of the GAS2-CH3 domain bound to F-actin provides the first example of the single CH3 domain interacting with F-actin.

The structures of CH1 bound to F-actin, such as those in FLNaCH1 (filamin) (Iwamoto et al, 2018), UTRN-ABD (utrophin) (Kumari et al, 2020), and T-plastin (Mei et al, 2022), have been previously elucidated by cryo-EM. A comparison of these CH1 domains with the CH3 domain bound to F-actin reveals significant differences. From the sequence similarity, we found the loops or short helices region of different type CH domains are diverse in length and sequence. First, the CH3 domain tends to have a longer and more well-ordered N-terminal (Appendix Fig. S3A green box). The longer N-terminal of the GAS2-CH3 domain forms a 'hook' structure to catch to the actin (Fig. EV5A, orange color represents GAS2-CH3 domain). Our biochemical study demonstrated that GAS2-CH3ΔN, which lacks the ABD1 region, significantly reduces its F-actin binding affinity (Fig. 2K). In contrast, cellular co-localization analysis by Kumari et al (2020) showed that the CH1 domain of utrophin, containing only ABD2' and ABD2, is sufficient for F-actin interaction in cells (Kumari et al, 2020). These findings suggest that ABD1 is crucial for GAS2-CH3's F-actin binding in vitro, but its necessity in cellular contexts requires further investigation.

From the sequence alignment, there is a rigid $3_{10}$ α-helix in the CH3 domain between helix E and F instead of the loops in the CH1 domain (Appendix Fig. S3A, red box). We also found the GAS2-CH3 domain is far away from the actin D-loop because of the rigid $3_{10}$ α-helix (Fig. EV5B, orange color represents GAS2-CH3 domain); this way would result in the weaker binding ability for GAS2-CH3 domain with actin. After structure alignment and measurement the footprint on actin for these CH domains (Liu et al, 2017), we found the T-plastin N-terminal does not interact with actin, FLNaCH1, UTRN-ABD, and GAS2-CH3 representatively interact with actin by N-terminal helix and loops (Fig. EV5A), especially, N-terminal of GAS2-CH3 domain has a larger

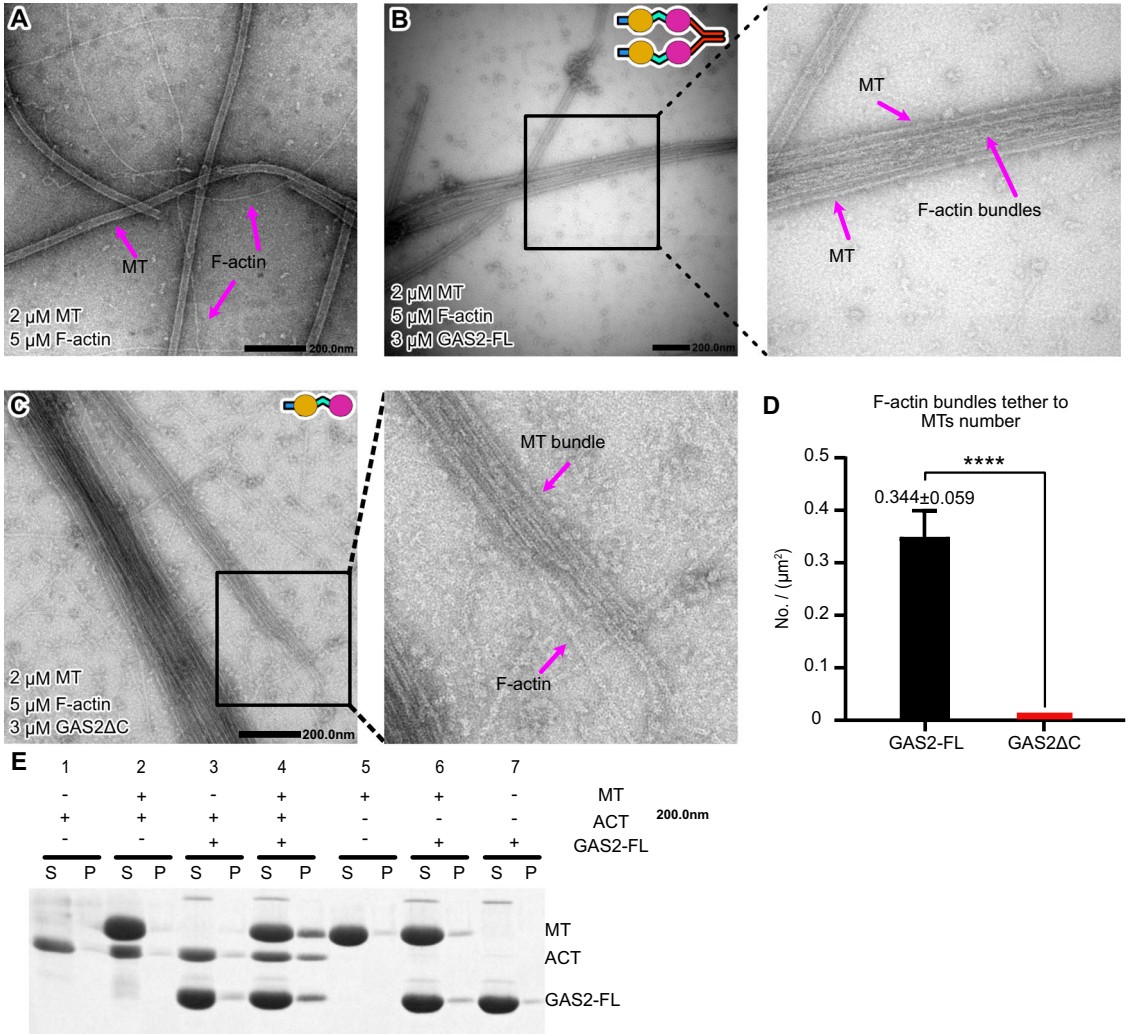

**Figure 7. GAS2 mediates F-actin and MT cross-linking.**

(A) A representative negative stain electron micrograph of mixture F-actin and MT (indicated by arrows), showing both cytoskeletons are observed as individual filaments. (B) A representative negative stain electron micrograph in the presence of GAS2-FL protein, F-actin, and MT. There are bundles of MT and F-actin in which large F-actin bundles are surrounded by MTs. (C) A representative negative stain electron micrographs in the presence of GAS2ΔC, F-actin, and MTs, F-actin was tethered to MT. (D) A histogram of the number of F-actin bundles tethering to MTs in the negative EM micrographs (mean ± SEM., from 16 independent views, 0.344 ± 0.059/μm² in the presence of GAS2-FL, 0/μm² in the presence of GAS2ΔC). ****$p < 1.0E-15$. (T-tests, nonparametric tests, two-tailed). $p$ values were calculated relative to the presence of the GAS2-FL condition. All the detailed values are shown in Source data. (E) A representative Coomassie-stained low-speed co-sedimentation assay gel containing supernatant (S) and pellet (P) for the mixture of MT, F-actin, and full-length GAS2. Quantitative analysis of the pellet is shown in Fig. EV4I–K. Source data are available online for this figure.

interaction area with actin (Fig. EV5F highlight with white circles). While FLNaCH1, UTRN-ABD, and T-plastin have a stronger interaction with actin at ABD2 by the flexible loops, the GAS2-CH3 domain has a rigid $3_{10}$ α-helix instead of the flexible loops, which decreases the binding area with actin (Appendix Figs. S3A and EV5C–F highlighted with a black box).

Our cryo-EM structure shows that the single CH3 domain is sufficient to bind with F-actin. The single EB-type CH domain exhibits a strong binding affinity for the microtubule plus-end and a lower binding affinity for F-actin, demonstrating its selectivity (Alberico et al, 2016). Based on the multiple sequence alignments, the EB-type CH domain lacks the ABD2' binding site, which explains its low affinity for F-actin (Appendix Fig. S4A brown box).

The EB-type CH domain also has a shorter loop between helix E and F, which is essential for CH1 and CH3 domains to contact with F-actin (Appendix Fig. S4A, red box). These results indicate that although the whole globular structure of the CH domain is conserved, the varying loops and short helix empower the diverse functions of the CH domain.

## GAS2 interacts with MT

The GAR domain is highly conserved from zebrafish to humans, especially for the microtubule-binding sites (β3-β4-β5-α2) and zinc-binding motif (Appendix Fig. S2A). This evolutionarily conserved amino acid sequence predicts the functional importance

of this domain. From our GAS2-GAR-MT complex structure, the GAS2-GAR domain specifically binds to the tubulin intradimer interface. Electrostatic analysis revealed complementary charges between the GAR domain and microtubules, with positively charged residues on the GAR domain aligning with negatively charged regions on tubulin intradimer (Fig. EV5G,H). The GAS2-GAR domain extends deeply into the intradimer concavity, interacting closely with β-tubulin's loop6-loop8 pocket, particularly aspartic acid 163 (β-D163), while α-tubulin's lysine 163 (α-K163) prevents binding due to steric hindrance and electrostatic repulsion (Figs. 4E and EV5G). To identify the GAS2-GAR domain binding similarity that may be involved in the interaction of tubulin with these motor proteins, we used the footprint method (Liu et al, 2017) to inspect potential contacts between tubulin and the binding partner. By comparison footprint on MTs, we observed that the GAS2-GAR domain, kinesin (PDB: 6WWI), and dynein (PDB: 6RZA) share a similar interaction surface on αβ-tubulin dimer, involving H11, H12 of α-tubulin and H4, H5, H12 of β-tubulin (Fig. EV5I).

In the presence of GAS2-FL or GAS2-GAR, tubulin rings were produced via the longitudinal growth of αβ-tubulin to form protofilament rings. This mechanism differs from the γ-TuRC, where γ-tubulins contact laterally to form a ring template. Upon incubation at 37 °C, these tubulin rings transitioned into sheets and microtubules over time, with a concomitant decrease in ring structures. This observation suggests that GAS2 facilitates tubulin ring formation, with these rings potentially serving as precursors for microtubule formation. However, we did not directly observe the incorporation of tubulin rings into sheets or microtubules, and we currently lack direct evidence for GAS2-GAR binding to free tubulin; such interactions could promote nucleation or affect the pool of polymerizable tubulin subunits; future studies should clarify this. Recent studies, such as Jijumon et al (2022), have shown that MACF1, a member of the spectraplakin family, promotes the formation of large tubulin ring structures with diameters of approximately 1 μm. These rings appear to serve as a template for microtubule nucleation, stabilizing tubulin dimers and facilitating microtubule growth. We observed a distinct phenomenon: GAS2-GAR induces the formation of much smaller ring-like structures (~50 nm in diameter). Unlike MACF1, these structures were not visible using TIRF microscopy, but we detected a gradual transition from spot-like aggregates to microtubule growth in the presence of GAS2-GAR, as shown in Movie EV9. This suggests that the mechanism by which GAS2-GAR influences microtubule ring structure may differ significantly from that of MACF1. In the presence of the GAS2-GAR domain, a large number of MTs were observed. However, with GAS2-FL, ring structures formed instead of MTs within 3 min. We hypothesize that the CH3 domain in the full-length GAS2 may inhibit the GAR domain's ability to bind to MTs. Removing the CH3 domain from GAS2 could release the GAR domain, enhancing its binding activity to MTs.

Our microtubule dynamics assay demonstrated that the GAS2-GAR domain stabilizes MTs by inducing a pausing state following polymerization. Interestingly, the GAR domain binds to regions on the MT lattice that overlap with those of motor proteins (Fig. EV5I). This finding suggests that the GAR domain may play a regulatory role akin to that of motor proteins and other microtubule-associated proteins. Zhang et al, (2018) reported that GMPCPP-stabilized MTs with 14 protofilaments exhibit a dimer rise of approximately 83.96 Å. Kinesin-1, known to stabilize MTs, reduces this spacing by 0.7 Å (Zhang et al, 2018). While in our study, we observed that GAS2-GAR-decorated MTs displayed a dimer rise of 82.25 Å, indicating a 1.7 Å reduction. This lattice compaction induced by GAS2-GAR likely contributes to the enhanced MT stability and the observed pausing of MT dynamics. Nevertheless, the influence of GAS2 on microtubule dynamics under cellular conditions remains elusive, and further investigations are needed to uncover the precise mechanisms of how the GAS2 protein regulates microtubule dynamics.

## GAS2 bundling and cross-linking ability in cell

A previous study demonstrated that Caspase-3 can cleave GAS2 at aspartic residue 279 during cell apoptosis, leading to cytoskeletal rearrangement, changes in cell morphology, and playing a crucial role in chondrogenesis (H. Lee et al, 1999; Sgorbissa et al, 1999). Our fluorescence intensity measurements and negative-stain microscopy reveal that the C-terminal region of GAS2 (residues 277–314) is essential for its F-actin bundling activity. Loss of this C-terminal region abolishes GAS2's ability to bundle F-actin, resulting in the disruption of F-actin integrity and the transition of F-actin into disordered individual filaments, ultimately causing cytoskeletal rearrangement during cellular processes (Fig. 8A). Additionally, our results show that GAS2 can tether F-actin bundles to microtubules. However, when GAS2 loses the C-terminal region, it loses its F-actin bundling ability. Although it obtains to tether single F-actin filaments to microtubule ability, the cross-linking pattern is altered (Fig. 7B,C). We hypothesize that this deletion imparts a different cross-linking ability to GAS2ΔC, which significantly disrupts cytoskeletal organization, causing cell membrane collapse and cell shape changes (Fig. 8A). To gain deeper insight into the structural basis of the GAS2 C-terminal region, we employed AlphaFold2 and AlphaFold3 to predict its dimerization based on the amino acid sequence from R271 to K314 (Jumper et al, 2021; Abramson et al, 2024). Both predictions suggest dimer formation of the C-terminal; however, AlphaFold2 predicts an α-helical structure with flexible loops (Appendix Fig. S5A), whereas AlphaFold3 suggests a β-sheet hairpin conformation (Appendix Fig. S5B). These theoretical models require further experimental validation to determine the physiologically relevant structure.

Our structural and biochemical data can explain how GAS2 would work in the inner ear. Pillar and Deiters' cells are specialized supporting cells with intricate morphologies and cytoskeletal specializations, forming strategic connections with inner and outer hair cells in the organ of Corti (Fig. EV1B) (Fettiplace and Hackney, 2006). In these cells, the microtubules are organized in a square array, interspersed with an equivalent number of actin filaments, with inter-microtubule distances ranging from 120 to 150 Å (Fig. EV1B) (Saito and Hama, 1982). GAS2 protein colocalizes with both actin filaments and microtubules in these supporting cells (Chen et al, 2021; Zhang et al, 2024). Our dark-field assays and negative-stain electron microscopy indicate that GAS2 can bundle microtubules into arrays in vitro (Fig. 3D–F). Our cryo-EM data demonstrates that the distance between microtubules ranges from 120 to 220 Å in the presence of GAS2 (Fig. 5D–F), consistent with observations in Pillar cells, suggesting a role for GAS2 in organizing microtubule distances. GAS2 may act

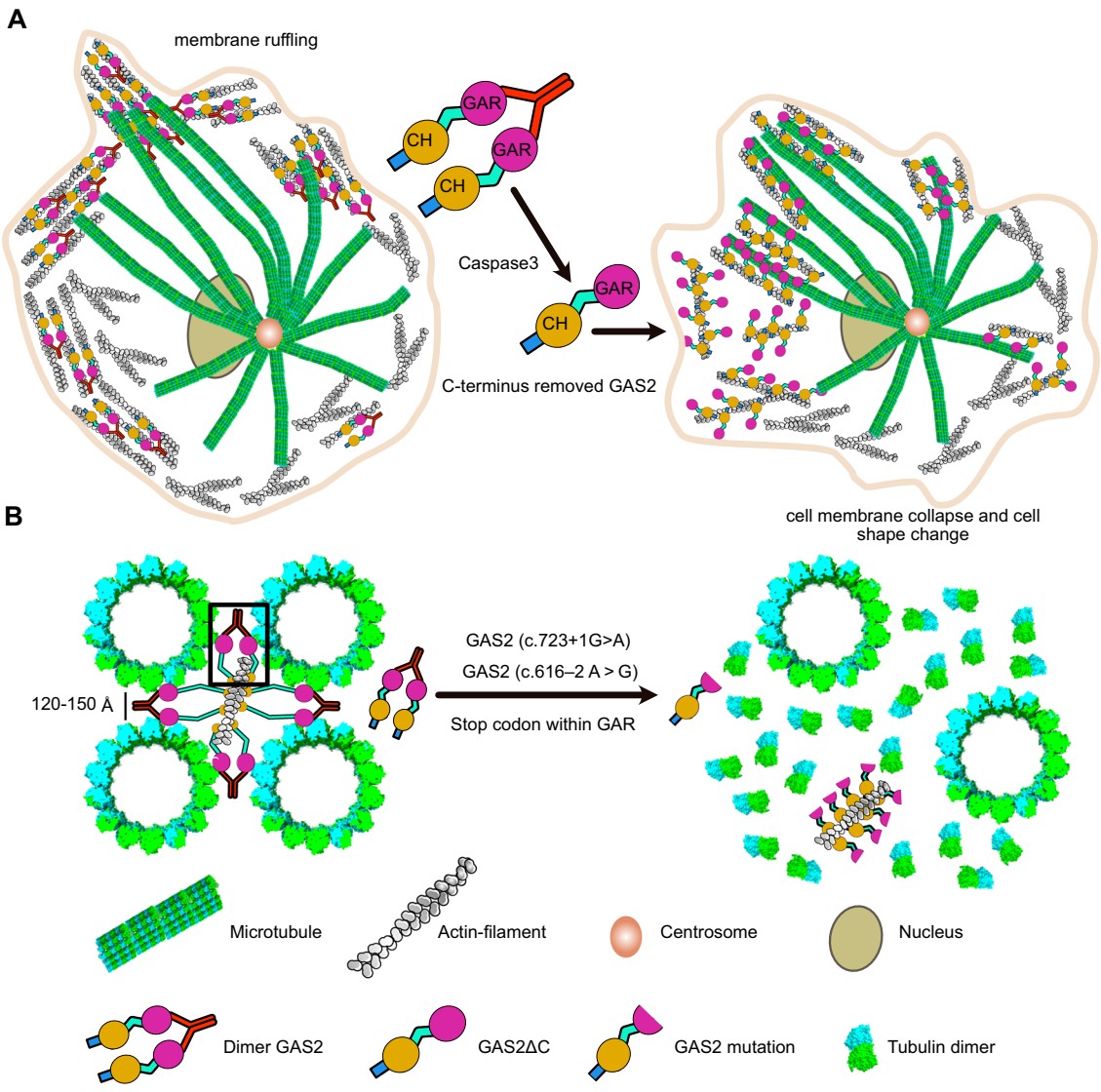

**Figure 8. Model of GAS2-driven microtubule bundling and F-actin bundling.**

(A) A simple work model for GAS2-FL in the cells. The GAS2-FL C-terminal being digested by caspase3, removed, or mutated would lead to disordered actin filament arrays or MT-arrays, causing the specifically differentiated cell shape changing or cell architecture catastrophe. (B) Schematic showing how GAS2 arranges MT and F-actin in Pillar/Deiters' cell. One GAS2 dimer may work like a flexible bridge, and four GAS2 dimers would bind with MT and link to F-actin to form a square. The black box shows one GAS2 dimer catches two MTs together by two GAR domains, and two CH3 domains tether to the F-actin. The mutations that disrupt the GAR domain and delete the C-terminal may cause GAS2 to lose its MT bundling and stabilization ability, resulting in MT depolymerization and MT array catastrophe.

as a flexible bridge in Pillar cells, eight GAR domains of four dimerization GAS2 pairs bundle four MTs at an optimal distance to form a square, and the CH3 domains tethering to F-actin, thus a perfect MTs-F-actin cross-linking arrays are formed (Fig. 8B, black box shows one of the GAS2-FL dimer units, Fig. EV1B).

Our biochemical data can explain the genetic diseases caused by GAS2 mutations. Variants such as GAS2 c.723+1 G > A and GAS2 c.616−2 A > G introduce a stop codon within the GAR domain, resulting in disordered microtubule arrays, microtubule depolymerization, and inherited hearing loss (Chen et al, 2021; Zhang et al, 2024). Our TIRF data shows that GAS2-GAR reduces catastrophe and promotes the rescue of MTs, stabilizes MTs.

However, the variants disrupting the GAR domain would impair GAS2's ability to stabilize MTs, leading to microtubule depolymerization. Microtubule bundle arrays are not formed when GAS2 lacks the C-terminal region (Fig. 3B), indicating a loss of MT bundling ability in Pillar and Deiters' cells. Variants such as GAS2 c.723+1 G > A and GAS2 c.616−2 A > G lead to C-terminal deletion and GAR domain disruption, causing GAS2 to lose its MT bundling ability, impairing stabilization, and resulting in MT depolymerization and MTs arrays catastrophe (Fig. 8B). Additionally, there are three non-centrosomal microtubule-organizing centers (nc-MTOCs) in Pillar cells generating MT bundle arrays (Fig. EV1B) (Henderson et al, 1995). The GAS2 C-terminal region is crucial for

ring structure formation. The truncation of this region results in the disappearance of ring structures and reduced microtubule polymerization compared to GAS2-FL or GAS2-GAR constructs. These results indicate that GAS2-FL potentially assists in nc-MTOC formation, and C-terminal deletion may also impair this function.

In summary, our research revealed that the GAS2 protein can promote MT nucleation, facilitate MT stabilization, bundle MTs or F-actin together, and crosslink F-actin bundles to MTs. Our research significantly enhances the molecular diagnosis of hearing loss and lays the groundwork for future studies to develop new therapeutic strategies to prevent progressive hearing loss linked to GAS2.

# Methods

### Reagents and tools table

| Reagent/Resource | Reference or Source | Identifier or Catalog Number |
|---|---|---|
| **Recombinant DNA** | | |
| pCold-Pros2 | Takara-bio | 3371 |
| Modified pCold-Pros2 | This study (Remove Thrombin and Factor Xa sites from pCold-Pros2 vector) | N/A |
| **Oligonucleotides and other sequence-based reagents** | | |
| PCR primers | This study | Primer dataset |
| **Chemicals, Enzymes and other reagents** | | |
| EcoRI-HF | https://www.neb.com/ja-jp/products/r3101-ecori-hf | R3101S |
| NdeI | https://www.neb.com/ja-jp/products/r0111-ndei | R0111S |
| DNA Polymerase/KOD plus Neo | https://www.toyobo-global.com/products/lifescience/products/pcr_016/index.html | KOD-201 |
| HEPES | https://labchem-wako.fujifilm.com/jp/product/detail/W01T02GB10.html | 342-01375 |
| Potassium Chloride | https://www.e-nacalai.jp/ec2/EC-srchdetl.cfm?jump=EC-srchdetl&syohin=2851475&syubetsu=3 | 28514-75 |
| HRV3C protease | https://catalog.takara-bio.co.jp/product/basic_info.php?unitid=U100007643 | 7360 |
| Polyoxyethylene(10) octylphenyl ether (Triton X-100) | https://labchem-wako.fujifilm.com/europe/product/result/product.html?fw=168-11805 | 168-11805 |
| Magnesium chloride hexahydrate | https://www.e-nacalai.jp/ec2/EC-srchdetl.cfm?jump=EC-srchdetl&syohin=2090955&syubetsu=3 | 20909-55 |
| EGTA | https://www.dojindo.com/products/G002/ | 346-01312 |
| (±)-Dithiothreitol | https://labchem-wako.fujifilm.com/jp/product/detail/W01W0104-0897.html | 045-08974 |
| Alexa Fluor 568 Phalloidin | https://www.thermofisher.com/order/catalog/product/A12380 | A12380 |
| PIPES | https://labchem-wako.fujifilm.com/jp/product/detail/W01T02GB15.html | 345-02225 |

| Reagent/Resource | Reference or Source | Identifier or Catalog Number |
|---|---|---|
| **Software** | | |
| ImageJ | https://imagej.nih.gov/ij/index.html | |
| GraphPad Prism | https://www.graphpad.com | |
| RELION | https://relion.readthedocs.io/en/release-5.0/ | |
| Pymol | https://www.pymol.org/ | |
| Chimera | https://www.cgl.ucsf.edu/chimera/ | |
| ChimeraX | https://www.cgl.ucsf.edu/chimerax/ | |
| VMD | https://www.ks.uiuc.edu/Research/vmd/ | |
| NAMD | https://www.ks.uiuc.edu/Research/namd/ | |
| Phenix | http://phenix-online.org/ | |
| **Other** | | |
| Quantifoil R1.2/1.3 Au 300 mesh grid | https://webcatalog.listill.com/products | X-101-Au300 |
| Vitrobot Mark IV System | Thermo Fisher Scientific | |
| ECLIPSE Ti2-E microscope | Nikon | |
| Titan Krios G3i microscope | Thermo Fisher Scientific | |
| FV3000 confocal microscope | Olympus Life Science | |
| Dark-field microscopy-BX53 | Olympus Life Science | |

## Cloning, expression, and recombinant protein purification

Mouse GAS2 constructs were subcloned into modified pCold-Pros2 vectors, where the Thrombin and Factor Xa sites had been removed. GAS2-FL (1–314), GAS2ΔC (1–276), GAS2-CH3 (1–201), GAS2-CH3ΔN (34–201), GAS2-GAR (198–314), GAS2-GARΔC (198–276) and their mutants were inserted into pCold-ProS2 after an HRV3C cleavage site (LGVLFG/GP). For GFP-tagged constructs, EGFP was inserted at the C-terminus of the protein within the pCold-ProS2 plasmid. The primers used for PCR amplification in construction of these plasmids are summarized in Table EV1. Proteins were expressed in BL21(DE3) at 20 °C for 16 h following induction with 0.5 mM IPTG at OD600 = 0.6. Bacteria were lysed by sonication in buffer A (30 mM HEPES, pH 7.5, 300 mM KCl). Lysates were centrifuged at $20,000 \times g$ for 30 min at 4 °C. Ni-NTA chromatography was used for protein purification. ProS2 tags were removed by on-column digestion with HRV3C protease in buffer A. GAS2-FL, GAS2ΔC, GAS2-GAR, GAS2-GARΔC, and their mutants were dialyzed against buffer B (25 mM Tris-HCl, pH 8.0, 150 mM KCl, 100 μM ZnCl₂, 1 mM DTT). GAS2-CH3, GAS2-CH3ΔN, and their mutants were dialyzed against Tris-Triton buffer (50 mM Tris-HCl, pH 8.0, 200 mM KCl, 2 mM MgCl₂, 1 mM DTT, 0.01% Triton X-100). GFP-tagged constructs

were purified using the same protocol as their non-GFP counterparts.

## Actin and tubulin preparation

Rabbit skeletal muscle actin was prepared from acetone powder (gift from Hirokawa lab) as described previously (Spudich and Watt, 1971). Briefly, actin was extracted from the acetone powder, polymerized with 0.1 M KCl, and then further purified with high-molar KCl (0.6 M) and depolymerization in the low salt buffer. F-actin was polymerized in F-buffer (50 mM Tris-HCl, pH 8.0, 200 mM KCl, 2 mM $MgCl_2$, 1 mM EGTA, 4 mM DTT). Tubulin was purified from porcine brains using a high-molarity PIPES buffer (Castoldi and Popov, 2003). GMPCPP-stabilized microtubules were prepared by polymerizing tubulin with GMPCPP as described by Yajima et al, (2012) (Yajima et al, 2012). Taxol-stabilized microtubules were prepared by polymerizing 5 mg/mL tubulin in the BRB-80 buffer (80 mM PIPES, pH 6.8, 2 mM $MgCl_2$, 1 mM EGTA, 1 mM DTT) with 50 mM GTP at 37 °C for 30 min, followed by the gradual addition of Taxol to a final concentration of 20 μM.

## Size exclusion chromatography (SEC) assays

SEC assays were performed using a Superdex 75 HR 10/300 column on an AKTA purifier system. GAS2-FL, GAS2ΔC, GAS2-GAR, and GAS2-GARΔC were loaded with SEC buffer E (30 mM Tris-HCl, pH 8.0, 300 mM KCl, 100 μM $ZnCl_2$, 0.1% β-ME). Eluted fractions were analyzed by SDS-PAGE. Standard proteins (BSA, 66 kDa; Carbonic Anhydrase, 29 kDa; Cytochrome c, 12.4 kDa) were used to generate a calibration curve, which was then used to estimate the molecular weights of GAS2-FL and its constructs based on their elution volumes (Fig. EV1C–E).

## Actin binding and bundling assays

High-speed co-sedimentation assays were performed to investigate the interaction between GAS2 constructs and F-actin. Various concentrations of GAS2 proteins were incubated with pre-polymerized F-actin (5 μM) in the F-buffer for 1 h at room temperature. The mixtures were then centrifuged at $186,000 \times g$ for 10 min at 25 °C. Supernatants and pellets were separated and analyzed using SDS-PAGE. The amount of GAS2 proteins in the pellet fractions was quantified using ImageJ software (NIH) to determine the binding affinity.

Low-speed co-sedimentation assays were conducted to assess the F-actin bundling activity of GAS2 constructs. F-actin (5 μM) was incubated with various concentrations of GAS2 proteins in the F-buffer for 1 h at room temperature. The mixtures were centrifuged at $9000 \times g$ for 10 min at 4 °C. Supernatants and pellets were separated and analyzed using SDS-PAGE. The percentage of actin in the pellet fractions was quantified to evaluate the bundling efficiency of each construct.

The F-actin bundles induced by GAS2 were visualized using fluorescence microscopy and negative stain electron microscopy. For fluorescence microscopy, F-actin (5 μM) was incubated with GAS2-FL (1, 3, and 5 μM) in an F-buffer for 1 h at room temperature. The mixture was diluted 10-fold with imaging buffer (50 mM Tris-HCl, pH 8.0, 200 mM KCl, 2 mM $MgCl_2$, 1 mM EGTA, 4 mM DTT, 50 μL/mL Alexa Fluor 568 phalloidin), applied to poly-L-lysine coated coverslips, and imaged using a FV3000 (Olympus Life Science) confocal microscope. For negative stain electron microscopy, the same procedure was followed as for fluorescence microscopy, except that the sample mixture was diluted with F-buffer and applied to glow-discharged carbon-coated copper grids, stained with 1% uranyl acetate, and imaged using a JEOL JEM-1400 transmission electron microscope operated at 100 kV.

## Microtubule binding and bundling assays

Co-pelleting assays were performed to confirm the binding affinity of GAS2-FL and its GAS2-GAR domain (198–314) to microtubules. Different concentrations of GAS2 proteins were incubated with Taxol-stabilized MTs in BRB80 buffer supplemented with 1 mM GTP at room temperature for 20 min. The mixtures were then centrifuged at $186,000 \times g$ for 10 min at 25 °C. All experiments were performed in triplicate, then supernatants and pellets were separated and analyzed by SDS-PAGE. The gels were quantified using ImageJ software (NIH), background subtraction was performed using the respective GAS2 construct controls, and statistical analysis was performed using GraphPad (GraphPad Software, San Diego, USA).

Dark-field microscopy and electron microscopy were used to observe MT bundling induced by GAS2 proteins. Taxol-stabilized MTs (2 μM) were incubated with 1.0 μM GAS2 proteins (GAS2-FL, GAS2ΔC, GAS2-GAR, GAS2-GARΔC) in BRB80 at room temperature for 30 min. The samples were then imaged using either dark-field microscopy (BX53, Olympus) or negative stain electron microscopy as described for the F-actin bundling assay above.

## Full-length GAS2/F-actin/MT cross-linking assay

Taxol-stabilized MTs and F-actin were prepared as described in the "Actin and tubulin preparation" section. 5 μM taxol-stabilized MTs, 5 μM F-actin, and 25 μM GAS2 proteins were mixed in BRB80 and incubated for 40 min at room temperature. After centrifugation at $800 \times g$ for 10 min, supernatants (S) and pellets (P) were separated and analyzed by SDS-PAGE. Control experiments included 5 μM bare MTs, 5 μM bare F-actin, 5 μM bare MTs mixed with 5 μM bare F-actin, 5 μM MTs mixed with 25 μM full-length GAS2, 5 μM F-actin mixed with 25 μM full-length GAS2, and 25 μM full-length GAS2 alone.

For negative stain electron microscopy to examine the F-actin/MT cross-linking ability of full-length GAS2, 2 μM MTs, and 5 μM polymerized actin filaments were incubated with 5 μM full-length GAS2 in BRB80 for 40 min at room temperature. The mixture was then diluted by adding 5 μL to 45 μL BRB80 buffer. Negative staining and imaging with electron microscopy were performed as described for the F-actin bundling assay above.

## Cryo-EM sample preparation and data collection

For GAS2-CH3-F-actin complex, 40 μL of Actin (4 μM polymerized in F-buffer) was mixed with 20 μL of 56 μM GAS2-CH3 at room temperature for 40 min in F-buffer, followed by the addition of another 20 μL of 56 μM GAS2-CH3 (residues 1–201) for further incubation for 40 min. 3 μL of the mixture was applied to a glow-

discharged carbon-coated grid (Quantifoil R1.2/1.3 Au300 mesh) and incubated for 60 s in the chamber of a Vitrobot (22 °C, 100% relative humidity; Thermo Fisher Scientific). The grid was blotted for 4 s at force 10 and plunged into ethane immediately.

For the GAS2-GAR-MT complex, 4 μL of 0.5 μM GMPCPP-microtubules were applied to a grid (Quantifoil R1.2/1.3 Au300 mesh) and incubated for 30 s in the chamber of a Vitrobot. Then 2 μL of 5 μM GAS2-GAR in BRB80 were added on the grid and incubated for 30 s to allow GAS2-GAR binding to the microtubule. 4 μL of the protein mixture was removed from the grid, and another 2 μL of GAS2-GAR was added. After another 30 s incubation, the grid was blotted for 3 s at force 15 and plunged into ethane immediately.

Images were recorded at 300 keV using a Titan Krios G3i microscope (Thermo Fisher Scientific). Images were collected with a Gatan-LS Energy Filter (Gatan, Pleasanton, CA) with a slit width of 20 eV and a Gatan K3 Summit direct electron detector in the electron counting mode. The magnification was set to 105,000× with a physical pixel size of 0.83 Å/pixel for the GAS2-CH3-F-actin complex or 0.83 Å/pixel for GAS2-GAR-MT. For the GAS2-CH3-F-actin complex, each movie was recorded with a total dose of 48.93 electrons/Å2 and subdivided into 49 frames. Movies were acquired using EPU software (Thermo Fisher Scientific), and the target defocus range was set from −2.4 to −1.2 μm. A total of 6378 micrographs were collected. For the MT-GAS2GAR complex, images were collected in the same condition as above. The total dose of 48.79 electrons/Å$^2$ was subdivided into 48 frames. Movies were acquired using SerialEM software (Mastronarde, 2005), and the target defocus range was set from −1.8 to −1 μm. A total of 3322 micrographs were collected.

## Image processing and 3D reconstruction

For both datasets, beam-induced motion correction was performed using MotionCor2 (Zheng et al, 2017) with 3 × 3 patches, and the CTF was estimated using CTFFIND4 (Rohou and Grigorieff, 2015).

For the GAS2-CH3-F-actin complex, 5756 micrographs were selected by subset selection in RELION (Scheres, 2012) with a maximum resolution cutoff of 5.5 Å determined from the CTF resolution estimation of the raw micrographs. In total, 29,620 segments were manually picked to avoid selecting F-actin filaments without GAS2-CH3 binding. 632,290 particles were extracted with a 27.4 Å helical rise. Several cycles of 2D classification and 3D classification were performed, and a final set of 412,176 particles was used for 3D model reconstruction. After polishing, the final resolution was 2.8 Å (Fig. EV2A).

For the MT-GAS2GAR complex, PyfilamentPicker scripts were used to select 14-PF MTs and cut each MT into overlapping boxes with an 82-Å rise (Zhang and Nogales, 2015; Nishida et al, 2020). All selected MTs were stacked into "super-particles". FREALIGN version 9 was used to determine the helical parameters (twist and rise) for the three-start helix of microtubules from the C1 reconstruction (Grigorieff, 2007; Lyumkis et al, 2013). Based on "super-particles" and seam search scripts, the seam location for each particle was determined after several cycles of FREALIGN local/global refinements. Poorly decorated microtubules and particles with incorrect seam locations were removed during processing. After iterative refinement, the seam location was visible in the C1 reconstruction. From the FREALIGN estimation, the resolution of the density map was 4.43 Å by 0.143 FSC value. Based

on this map, we fitted the atomic structure of the tubulin dimer and NMR structure of the GAS2-GAR (PDB: 1V5R) domain into the map and produced an initial atomic model. We copied this atomic model, filled all these models into the 3D density map, and then created 5 Å reference maps (Fig. EV3C) by Chimera (UCSF, https://www.cgl.ucsf.edu/chimera/) for the subsequent step analysis.

To further improve the resolution, the data was processed following the MiRP method based on the RELION pipeline (Cook et al, 2020) (Fig. EV3C). We used the same dataset and manually picked MT filaments by RELION v3.1, and then 4× binned particles were extracted. MT particles were assigned to 11–16 protofilament microtubules by 3D classification in RELION, and 14 protofilament MTs occupying 90% of the total picked MT were chosen for further reconstruction. Reference maps rebuilt from the FREALIGN refinement described above were used for initial seam alignment and seam checking. The final resolution and the B-factor for map sharpening were estimated using relion_post-process in RELION-3.1 (Table 1). Due to the presence of the seam, the MT reconstruction contains only one "good" protofilament at the opposite seam position (Zhang and Nogales, 2015; Cook et al, 2020), which was chosen for atomic model building.

For high-resolution analysis of the MT-GAS2-GAR complex, the images were reprocessed using CryoSPARC (version 4.62) (Punjani et al, 2017). Imported movies were first processed using patch motion correction and patch CTF estimation. Filaments were manually picked and segmented into 540 × 540 pixel boxes at an 82-angstrom step, generating approximately 1000 particles. Particles were 2D classified, good particles were selected and used with the Filament Tracer tool to pick additional filaments. The picked filaments were segmented at an 82-angstrom pitch, binned fourfold into 200 × 200 pixel boxes, and subjected to 2D classification to remove non-microtubule particles. After several rounds of 2D classification for cleanup, the particles were subjected to heterogeneous refinement using an 11–16 protofilament microtubule reference, also used for MiRP2 analysis, for further classification. Following several rounds of classification, the major homogeneous particles corresponding to 14-protofilament microtubules were selected and re-extracted at 1.0375 Å/pixel in 400 × 400 pixel boxes. Particles were refined using Helical Refinement without symmetry, resulting in a 3.7 Å resolution map. Subsequently, symmetry expansion was applied with a helical twist of −25.75° and a helical rise of 8.88 Å, based on the MiRP2 analysis of the 14-protofilament MT-GAS2-GAR. After symmetry expansion, particles were subtracted using a mask corresponding to two tubulin heterodimers with one GAS2-GAR domain generated from the MiRP2-derived model and a soft mask created with Relion 5.0. The subtracted particles were then refined using Local Refinement with non-uniform refinement and the same focus mask used for particle subtraction, resulting in the final map. Before final refinement, we conducted 3D classification; however, the classified classes did not show significant differences, so this strategy was not adopted. The global resolution of the composite maps was estimated using gold-standard Fourier shell correlation (FSC = 0.143), and the local resolution was determined using the local resolution estimation tool of CryoSPARC. Image processing and map quality are summarized in Appendix Fig. S1A.

## Model building and refinement

For the GAS2-CH3-F-actin complex, an atomic model was constructed using the AlphaFold2 (Jumper et al, 2021) prediction of the GAS2-CH3 domain (AF2 model No. AF-P11862-F1) as a starting

model fitted to density map by Chimera X (https://www.rbvi.ucsf.edu/chimerax/). Most of the GAS2-CH3 domain fit well into the density map, except for a few flexible loops. Initial atomic coordinates were generated and refined via molecular dynamics flexible fitting (MDFF) (Trabuco et al, 2008, 2009) to optimize the fit to the cryo-EM density map. The detailed procedure for MDFF is described below. Manual adjustments were made in Coot to improve the fit. Further refinements were performed using the real-space refinement tool in Phenix (Adams et al, 2010) and ChimeraX ISOLDE software (Pettersen et al, 2020). The atomic model was validated using the validation and map-based comparison tools in Phenix. The statistics of the final atomic models are summarized in Table 1. All the figures were prepared by PyMol (https://pymol.org/), Chimera or ChimeraX.

For the GAS2-GAR-MT complex, a single unit of tubulin dimer/GAR complex was extracted from the original density map. The NMR structure of the GAS2-GAR domain (PDB: 1V5R) and tubulin dimer structure (PDB: 3JAT) were rigidly fitted into the density map using ChimeraX. This combined atomic model is used as the initial one. After the Phenix refinement, we manually refined the structure with UCSF ChimeraX and ISOLDE software, using MiRP and CryoSPARC density maps simultaneously. Finally, the atomic model was validated using the validation and map-based comparison tools in Phenix.

### Molecular dynamics simulation (MDFF)

The rigid fitted initial atomic model (GAS2-CH3-F-actin or GAS2-GAR-MT) was loaded into VMD software to prepare the PSF file. Then, a solvation box was added to the model. Following the next step, the model was ionized with neutralized NaCl ions. After that, the model was performed MDFF by using NAMD software; 50,000 steps were used with 0.3 kcal/mol force-scaling factor for cryo-EM density map and with the presence of secondary structure restraints and cis-peptide restraints. Then, we performed the energy minimization of the model for 2000 steps using a force-scaling factor of 10.0 kcal/mol.

### Microtubule polarity trace

The polarity of individual microtubules was determined by matching the 3D structure reconstruction of the GAS2-GAR-MT complex to the image. First, the polarity of the final GAS2-GAR-MT was established by using ChimeraX based on the known microtubule structure. Then, we assign the polarity to individual particles in the 2D raw micrographs based on the tilt and ψ angle data stored in the RELION Star file. This polarity information was then used to overlay arrows onto the raw micrographs, where the arrow orientation reflects the polarity from the microtubule minus end to the plus end. Refer to the Code EV file for detailed script information.

### Time-dependent microtubule nucleation assay by electron microscopy

Time-dependent microtubule nucleation assay by EM as described previously (Wang et al, 2014; Imasaki et al, 2022). Briefly, 3 μM tubulin was incubated with 3 μM GA2-FL, or GAS2-GAR, or GAS-FLΔC, or GAS2-GARΔC for 10 min on ice in BRB80 buffer. Then, the mixture was transferred to 37 °C for 1, 3, 10, or 30 min. After incubation, 3 μl of each sample was applied to glow-discharged

carbon-coated copper grids, stained with 1% uranyl acetate, and imaged using a JEOL JEM-1400 transmission electron microscope operated at 100 kV.

### MT dynamic assay

In vitro MT dynamics were observed under the TIRF microscope as described previously (Taguchi et al, 2022). The flow chamber was assembled using microscopy slides and precleaned glass coverslips with double-sided tape. The chamber was treated with 0.5 mg/mL PLL-PEG-biotin in BRB80 buffer for 5 min, washed with BRB80 buffer, and then incubated with 0.5 mg/mL streptavidin for 5 min. Short MT seeds were prepared using a 1.5 μM tubulin mix containing 50% biotin-tubulin and 50% AZdye647-tubulin with 1 mM GMPCPP at 37 °C for 30 min. The MT seeds were attached to the coverslips using biotin-avidin links and incubated with an assay buffer.

The reaction mixture containing a specified amount of GAS2-GAR-GFP (100 nM), 10 μM tubulin, 1 mM GTP, an oxygen scavenging system composed of Trolox/PCD/PCA, and 2 mM ATP was added to the flow chamber. The samples were maintained at 30 °C during the experiments. An ECLIPSE Ti2-E microscope equipped with a CFI Apochromat TIRF 100XC Oil objective lens, an Andor iXion Life 897 camera, and a Ti2-LAPP illumination system (Nikon, Tokyo, Japan) was used to observe single-molecule motility. NIS-Elements AR software ver. 5.2 (Nikon) was used to collect movie data for 20 min.

## Data availability

Our cryo-EM density maps and atomic coordinates were deposited in the Electron Microscopy Data Bank (EMDB) and the Protein Data Bank (PDB) with the following accession codes: GAS2-CH3-F-actin (PDB ID: 9KBX, EMDB: EMD-62233); GAS2-GAR-MT (PDB ID: 9KBW, EMDB: EMD-62232).

The source data of this paper are collected in the following database record: biostudies:S-SCDT-10_1038-S44318-025-00415-2.

## Peer review information

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

## Acknowledgements

This work was supported by JSPS KAKENHI Grant Number 21H04762 and 21H05248 to MK, 21H05254 to RN, 22H05523 to SN, and 23K27145 to AN. This work was supported by the Platform Project for Supporting Drug Discovery and Life Science Research (Basis for Supporting Innovative Drug Discovery and Life Science Research (BINDS)) from AMED under Grant Number JP22ama121002j002 (to MK). Figure EV1B was drawn using images from Servier Medical Art. Servier Medical Art by Servier is licensed under a CC BY 4.0 Unported License (https://creativecommons.org/licenses/by/4.0/).

## Author contributions

**Jiancheng An**: Conceptualization; Data curation; Formal analysis; Validation; Investigation; Visualization; Methodology; Writing—original draft; Writing—review and editing. **Tsuyoshi Imasaki**: Data curation; Formal analysis; Supervision; Validation; Investigation; Methodology; Writing—original draft; Writing—review and editing. **Akihiro Narita**: Resources; Formal analysis; Funding acquisition; Validation. **Shinsuke Niwa**: Resources; Formal analysis; Funding acquisition; Methodology. **Ryohei Sasaki**: Methodology. **Tsukasa Makino**: Formal analysis; Supervision; Investigation; Visualization; Methodology. **Ryo Nitta**: Resources; Formal analysis; Supervision; Funding acquisition. **Masahide Kikkawa**: Resources; Formal analysis; Supervision; Funding acquisition; Validation; Investigation; Writing—original draft; Project administration; Writing—review and editing.

Source data underlying figure panels in this paper may have individual authorship assigned. Where available, figure panel/source data authorship is listed in the following database record: biostudies:S-SCDT-10_1038-S44318-025-00415-2.

## Disclosure and competing interests statement

The authors declare no competing interests.

# Expanded View Figures

**Figure EV1.  Function of GAS2 protein in the cochlea, its purification, and assessment of its F-actin bundling activity.**

(**A**) Domain structure of spectraplakins. (**B**) The schematic diagram illustrates the inner ear. Sound waves entering the outer ear vibrate the eardrum, with these vibrations amplified by the ossicles (malleus, incus, and stapes) and transferred to the oval window. The oval window transmits the vibrations to the cochlea, where fluid movement creates waves that cause the basilar membrane to vibrate (left panel). These vibrations are transduced into inner and outer hair cells (IHC, OHC) to generate an electrical signal supported by Pillar and Deiters' cells (middle panel). Pillar and Deiters' cells contain a dense cytoskeleton network; microtubules in tightly bundled square arrays spaced 120–150 Å apart and cross-linked with F-actin (left panel). This intricate structure is vital for sound wave oscillation by providing structural rigidity to the supporting cells. Three ncMTOCs (non-Centrosomal Microtubule-Organizing Centers) collaborate to coordinate control of microtubule positioning in the Pillar cells (middle panel). (**C**) SEC (Size Exclusion Chromatography) results for GAS2-FL, solid red line; GAS2ΔC, solid purple line; GAS2-GAR domain, solid green line; and GAS2-GARΔC, solid blue line with marker protein (black dot line). (**D**) The calibration formula for Superdex 75 HR 10/300 column according to the peak location of the marker proteins. (**E**) Calibration standard M.W. line for Superdex 75 HR 10/300 column according to peak location of marker proteins, shown as black dots and pink lines. The blue dot lines show the relation between the peak location and the molecular weight derived from the calibration standard line. (**F**, **G**) Representative Coomassie Blue-stained gels of co-sedimentation assays containing supernatant (S) and pellet (P) for purified GAS2-FL and GAS2-CH3 domain. (**H**, **I**) Low-speed co-sedimentation assay of purified full-length GAS2 (GAS2-FL) and C-terminal truncation GAS2ΔC. Various amounts of full-length GAS2 or GAS2ΔC were mixed with F-actin, incubated for 60 min at room temperature, and then centrifuged for 10 min at 9000 × *g*. Both supernatant (S) and pellet (P) were separated on SDS/PAGE and stained with CBB. All the raw SDS-PAGE gels are shown in Source data. Source data are available online for this figure.

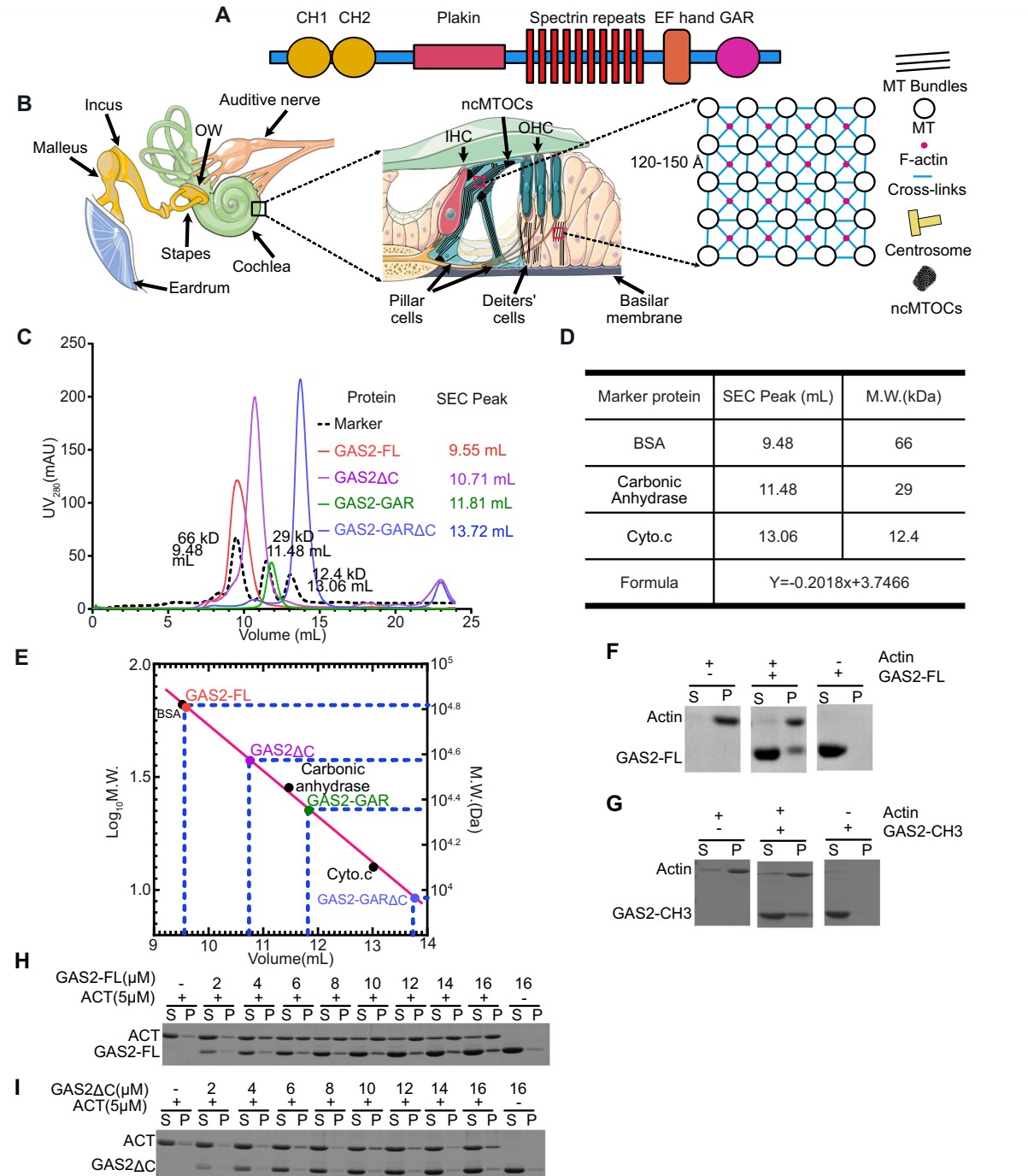

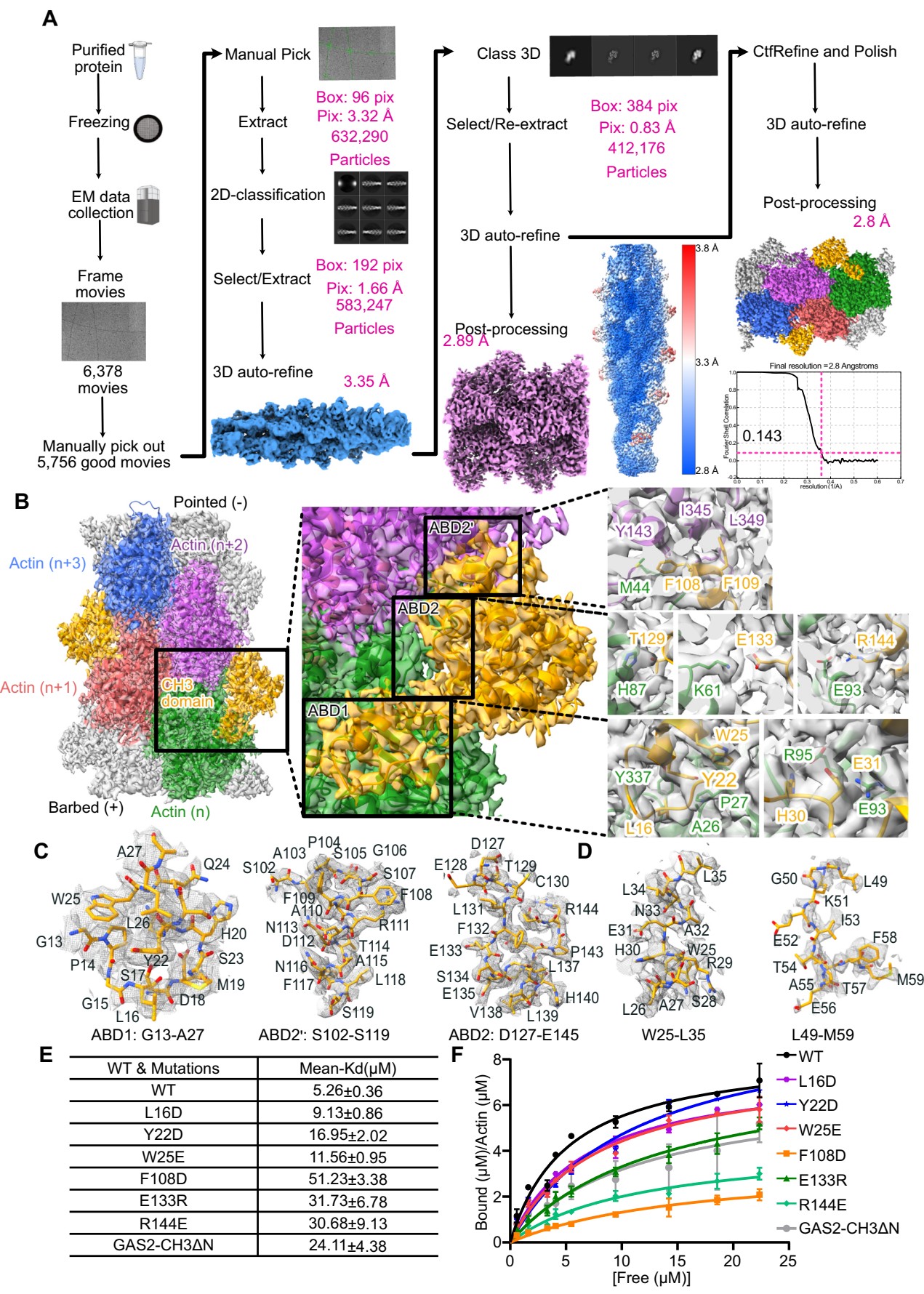

© The Author(s)

**Figure EV2.  Structural analysis of F-actin-GAS2-CH3 complex.**

(A) GAS2-CH3-F-actin structure reconstruction workflow by RELION v4.0. (B) The atomic models for the GAS2-CH3 domain fitted into the 3D reconstruction density map and residual level information of key amino acids interacting with actin with density map same as (Fig. 2D). Actin subunits (marked as n series from bottom to top) are presented in green, Indian red, medium orchid, and royal blue. The GAS2-CH3 domain is presented in orange. (C) Model-map overlaps in good fitting examples for GAS2-CH3-F-actin complex, including ABD1: G13-A27; ABD2': S102-S119; ABD2: D127-E145. (D) Model-map overlaps in bad-fitting examples for the GAS2-CH3-F-actin complex, including W25-L35 and L49-M59 peptides. Some density volumes are discontinuous. (E) The $K_d$ value for WT and mutants of the GAS2-CH3 domain according to co-pellet assay. Each $K_d$ value represents the mean ± SD. from three independent experiments. (F) The fitting curve for WT and mutants of the GAS2-CH3 domain for co-pellet assay. One site-specific binding method was ustoffitting the curve. Each data point represents the mean ± SD from three independent experiments. All the raw SDS-PAGE gels are shown in Source data.

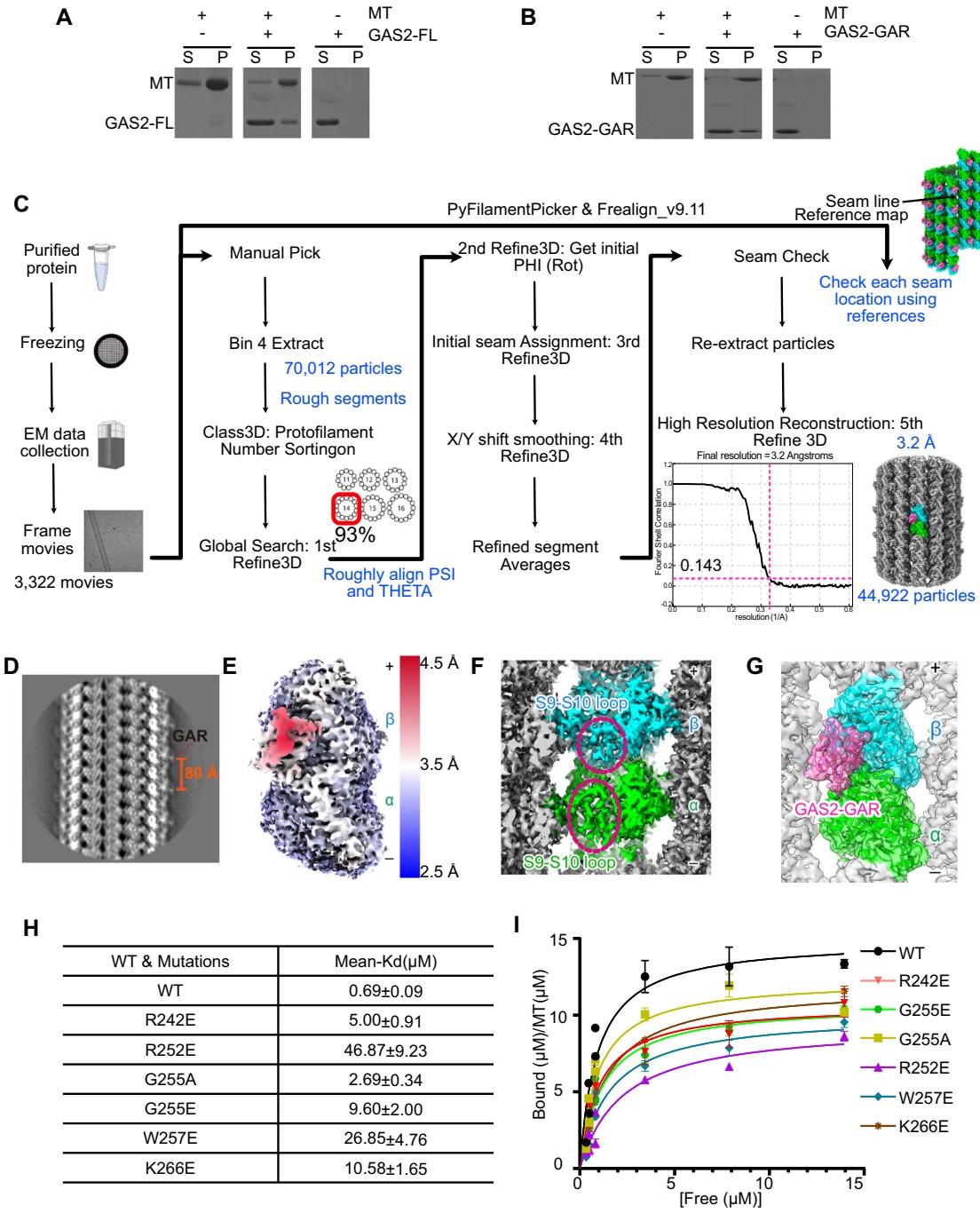

**Figure EV3. Structural analysis of MT-GAS2-GAR complex.**

(A, B) Representative Coomassie-stained MT co-sedimentation assay gels containing supernatant (S) and pellet (P) for purified GAS2-FL and GAS2-GAR domain. (C) GAS2-GAR-MT structure rebuilding workflow by MiRP. (D) The 2D class average for a GAS2-GAR decorated on a microtubule. GAS2-GAR domains are regularly spaced at 8 nm. (E) The local resolution estimation of the asymmetric units by RELION v3.1.3 for MiRP method reconstructed density map. Blue represents 2.5 Å, white represents 3.5 Å, red represents 4.5 Å. (F) Zoomed-in view of the cryo-EM reconstruction from the lumen for MiRP method reconstructed GAS2-GAR-MT density map. Red dashed circles highlight the density regions corresponding to the S9-S10 loop, which has different lengths in α- and β-tubulin (α-tubulin has a longer S9-S10 loop), aiding in subunit assignment. (G) Outside top view of atomic models for GAS2-GAR domain fitted in the 3D reconstruction density map by MiRP (α-tubulin in green, β-tubulin in cyan, GAS2-GAR domain in hot-pink). (H) The $K_d$ value for WT and mutants of the GAS2-GAR domain according to co-pellet assay. Each $K_d$ value represents the mean ± SD from three independent experiments. (I) The fitting curve for WT and mutants of the GAS2-GAR domain for co-pellet assay. One site-specific binding method was used to fit the curve. Each data point represents the mean ± SD from three independent experiments. All the raw SDS-PAGE gels are shown in Source data. Source data are available online for this figure.

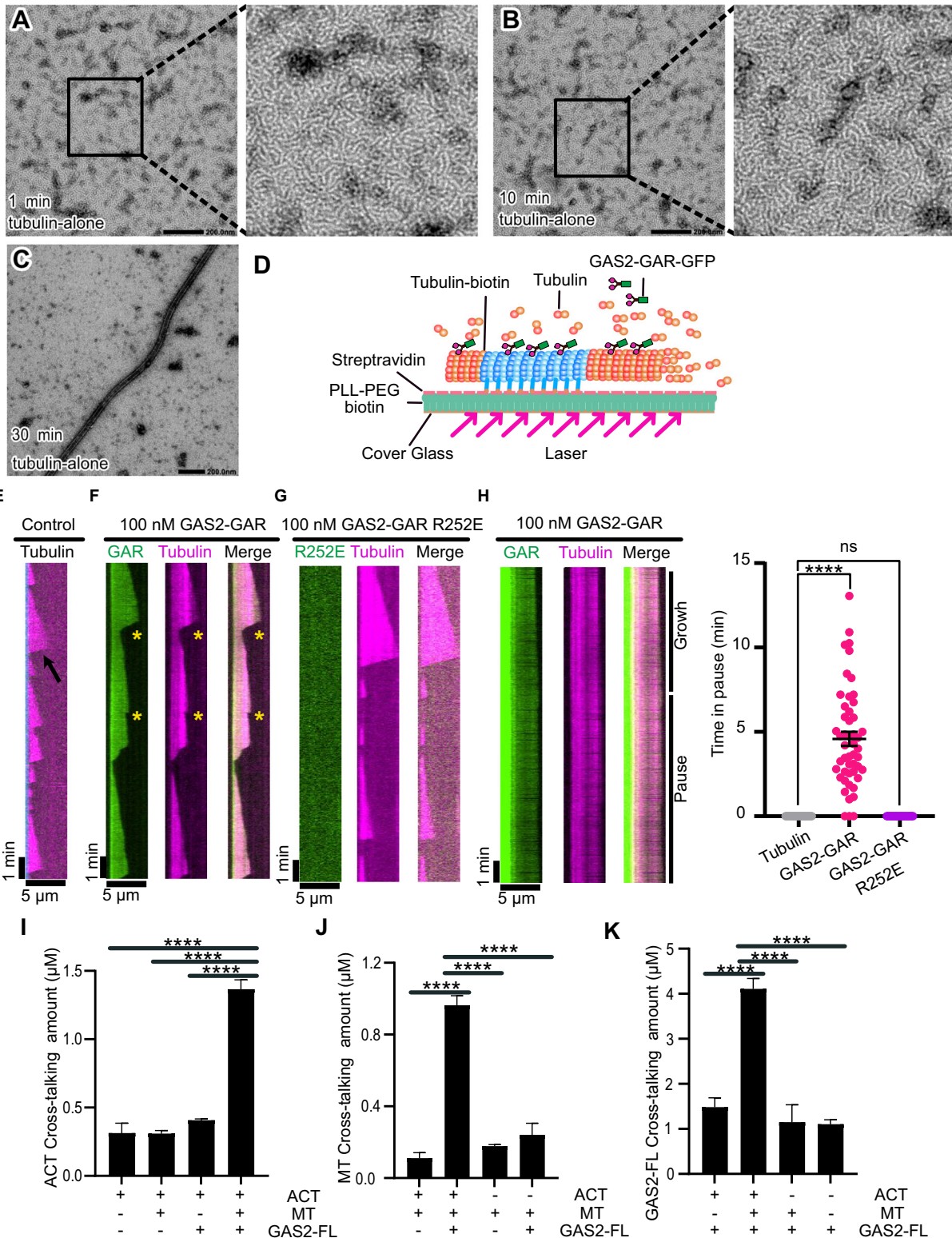

◀ **Figure EV4. The GAS2 protein facilitates microtubule nucleation and stabilization while also crosslinking F-actin and microtubules.**

(A–C) Representative negative stain EM micrographs of 3 μM tubulin-alone polymerization at different time points. (D) Schematic of TIRF microscopy-based MT stabilization assay. The biotin-labeled microtubule seed (cyan) was tethered to the cover glass by streptavidin and PLL-PEG biotin. The free tubulin dimers (shown in red and orange color) polymerize or depolymerize at the end of the microtubule seed. (E) A representative kymograph of microtubules (magenta) growing from GMPCPP seeds (cyan) in tubulin-alone. The black arrow shows one of the catastrophe events. (F) Representative kymographs of one microtubule (magenta) growing in the presence of GAS2-GAR-GFP (green). Yellow stars represent the rescue event. From left to right, the panels show the GAS2-GAR-GFP, tubulin (magenta), and the merged kymographs. (G) Representative kymographs of one microtubule (magenta) growing in the presence of GAS2-GAR-R252E-GFP (green). From left to right, the panels show the GAS2-GAR-R252E-GFP, tubulin (magenta), and the merged kymographs. (H) The left panel shows the representative kymograph of the pause state of microtubule dynamics in the presence of GAS2-GAR-GFP. "Growth" represents MT polymerization. "Pause" represents the MT pause state. The right panel shows the average pause time per microtubule. The black bars show mean ± SEM. For control, bare-MTs ($N = 19$) were analyzed. For GAS2-GAR-GFP, $N = 50$ MTs were analyzed. For GAS2-GAR-R252E-GFP, $N = 21$ MTs were analyzed. One-way ANOVA with Dunnett's multiple comparisons test has been used. **** $p$-value < 0.0001. $p$ values were calculated relative to the tubulin-alone condition. $p$-value from left to right for pause events:4.6E-11, 0.99. For all kymographs, horizontal scale bar = 5 μm; vertical scale bar = 1 min. (I–K) Histogram of pellet quantitative analysis for F-actin, MT, and GAS2-FL in (Fig. 7E). Each value represents the mean ± SD. from three independent experiments. One-way ANOVA with Dunnett's multiple comparisons test has been used. ****$p$-value < 0.0001. $p$-value in (I) from left to right: 2.4E−8, 2.3E−8, 5.0E−8; $p$-value in (J) from left to right: 3.8E−8, 7.3E−8, 1.4E−6; $p$-value in (K) from left to right: 4.1E−6, 1.6E−6, 4.1E−6.

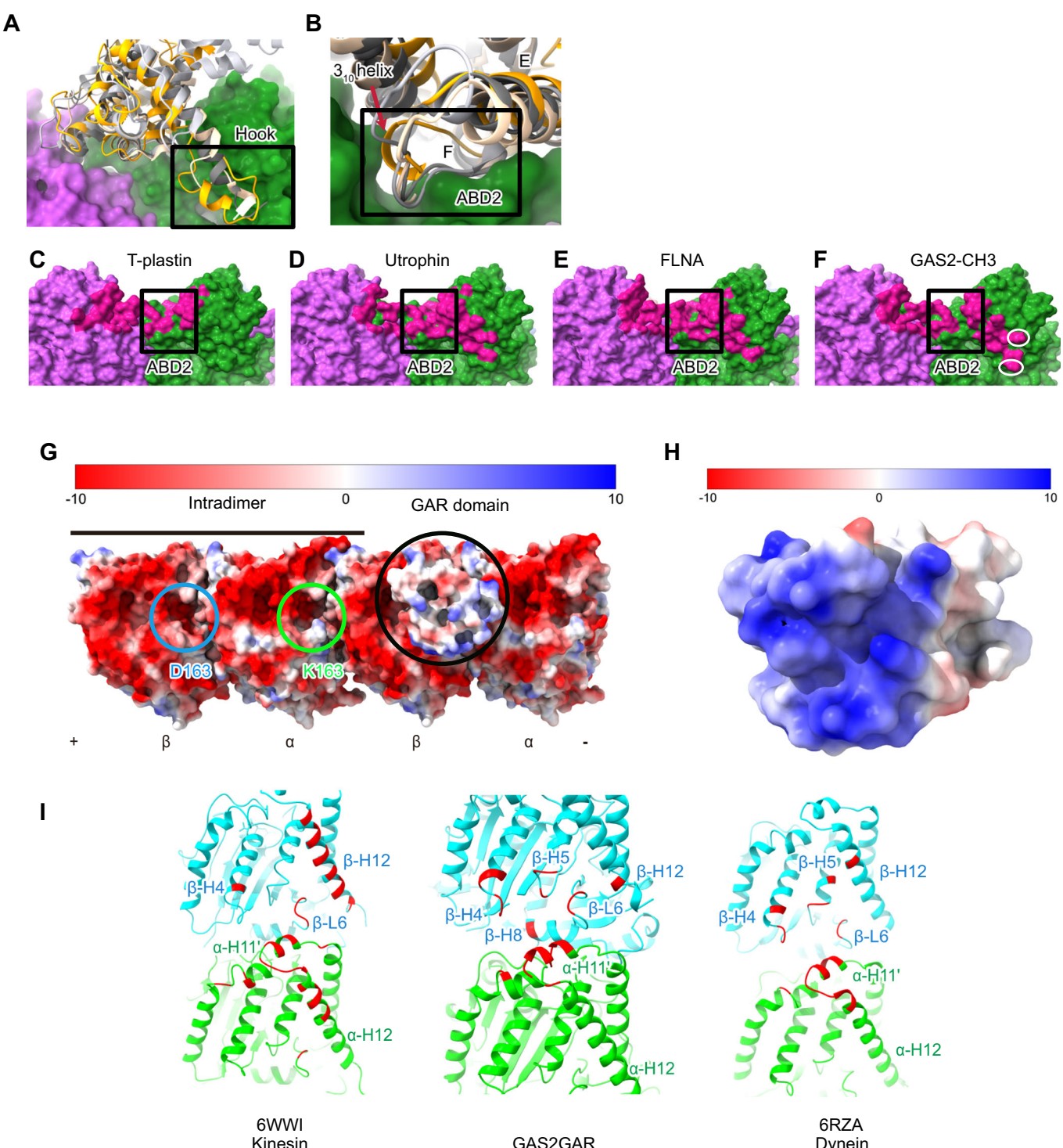

**Figure EV5.  Structural comparison of CH domains and electrostatic interactions analysis of GAS2-GAR with microtubules.**

(**A**) CH domain N-terminal structure alignment, GAS2-CH3 (orange) interacts with actin by a longer N-terminal compared with T-plastin (silver, PDB:7R94, EMDB: EMD-24323), utrophin (gray, EMDB: EMD-30085, PDB: 6M5G) and FLNA (wheat, EMDB: EMD-7831, PDB:6D8C). (**B**) GAS2-CH3 domain rigid 3₁₀ α-helix structure compares with the loops of T-plastin, utrophin, and FLNA by the same color scheme and structure data bank source as in (**A**). (**C–F**) Footprint method to inspect potential contacts between actin and the binding partner as defined within 5-Å distance by ChimeraX with the same structure data bank source as in (**A**). (**G**) Overall analysis of the electrostatic potential of the contact surfaces showed complementary charges between the GAS2-GAR domain and the MT. The black circle shows the GAS2-GAR domain well bound to charge complementary concave of the intra-tubulin dimer. The cyan box shows negatively charged amino acid D163 in β-tubulin. The lime box shows the positively charged amino acid D163 in α-tubulin. (**H**) The bottom of the GAR domain is full of positively charged amino acids. (**I**) Footprint method to inspect potential contacts between tubulin and the binding partner as defined within 5-Å distance by ChimeraX, Kinesin PDB:6WWI, and Dynein PDB: 6RZA were used.

