## [Peer Review File · The EMBO Journal]

Dimerization of GAS2 mediates crosslinking of microtubules and F-actin

Jiancheng An, Tsuyoshi Imasaki, Akihiro Narita, Shinsuke Niwa, Ryohei Sasaki, Tsukasa Makino, Ryo Nitta, and Masahide Kikkawa

Corresponding author(s): Jiancheng An (ankensei09@m.u-tokyo.ac.jp)

Review Timeline:

Submission Date:	12th Sep 24
Editorial Decision:	9th Oct 24
Revision Received:	23rd Jan 25
Editorial Decision:	20th Feb 25
Revision Received:	5th Mar 25
Accepted:	12th Mar 25

Editor: Ieva Gailite

Transaction Report:

Dear Dr. Kikkawa,

Thank you for submitting your manuscript for consideration by the EMBO Journal. We have now received comments from three reviewers, which are included below for your information.

As you will see from the reports, all reviewers appreciate the study and find it of broad interest to the research field. However, they also indicate substantive concerns regarding the structural analysis (reviewer #2) and request further analysis of the effect of GAS2 and its GAR domain on microtubule dynamics, as highlighted by reviewer #1 (points 10-11) and reviewer #3 (points 1-4).

Based on overall interest expressed in the reviewer reports, I invite you to revise the manuscript along the lines indicated in the referee comments, especially focusing on the points above. I think it would be helpful to discuss the revision in more detail via email or phone/videoconferencing - please let me know which option you prefer. I should also add that it is The EMBO Journal policy to allow only a single major round of revision and that it is therefore important to resolve the main concerns at this stage.

We generally allow three months as standard revision time. As a matter of policy, competing manuscripts published during this period will not negatively impact on our assessment of the conceptual advance presented by your study. However, please contact me as soon as possible upon publication of any related work to discuss the appropriate course of action. Should you foresee a problem in meeting this deadline, please let us know in advance to discuss an extension.

When preparing your letter of response to the referees' comments, please bear in mind that this will form part of the Review Process File and will therefore be available online to the community. For more details on our Transparent Editorial Process, please visit our website: <https://www.embopress.org/page/journal/14602075/authorguide#transparentprocess>. Please also see the attached instructions for further guidelines on preparation of the revised manuscript.

Please feel free to contact me if you have any further questions regarding the revision. Thank you for the opportunity to consider your work for publication. I look forward to discussing your revision.

Yours sincerely,

Ieva Gailite

Ieva Gailite, PhD
Senior Scientific Editor
The EMBO Journal
Meyerohofstrasse 1
D-69117 Heidelberg
Tel: +4962218891309
i.gailite@embojournal.org

- a point-by-point response to the referees' comments, with a detailed description of the changes made (as a word file).
- a word file of the manuscript text.
- individual production quality figure files (one file per figure)
- a complete author checklist, which you can download from our author guidelines

(<https://www.embopress.org/page/journal/14602075/authorguide>).

- Expanded View files (replacing Supplementary Information)

- a Reagents and Tools Table as part of the Methods section, which can be downloaded from our author guidelines

(<https://www.embopress.org/page/journal/14602075/authorguide#structuredmethods>)

We realize that it is difficult to revise to a specific deadline. In the interest of protecting the conceptual advance provided by the work, we recommend a revision within 3 months (7th Jan 2025). Please discuss the revision progress ahead of this time with the editor if you require more time to complete the revisions.

Referee #1:

This manuscript, "Dimerization GAS2 mediates microtubule and F-actin crosslinking," by An et al., reports the structure of GAS2 domains bound to actin and microtubule filaments. The structure-function work is of very good quality, and the manuscript attempts to present a model of GAS2-mediated crosslinking using various structural and biochemical methods. Few studies have reported F-actin and microtubule crosslinkers' structural and biochemical properties. Therefore, this manuscript will be of broad interest to the cytoskeleton and cell biology community. I have a few suggestions and recommendations for improving the manuscript before publication.

1. The authors have described the hearing loss mutations and correlated them with their biochemical work in the discussion and the model figure. It will be impactful if they include a sentence about this in their abstract.
2. In Figure 1A and Appendix Figure S1, the phosphorylation and caspase cleavage sites and the inherited hearing loss mutations can be marked.
3. The GAS2-CH3 domain boundaries used for cryoEM work need to be mentioned in the methods and the beginning of results describing cryoEM work.
4. In Figure 2 and Appendix Figure S2, the polarity of the F-actin should be marked.
5. Similarly, in Appendix Figure S2 B, the electron maps of critical interactions, as shown in Figure 2 D, can be shown instead.
6. The structure-function analysis of GAS2-CH3 binding to F-actin shows that ABD1 seems dispensable, similar to the UTRN studies by Kumari et al. However, the authors have mentioned that the hydrophobic pocket of ABD1 is critical. This needs some reconciliation between their results and conclusions.
7. In the cryoEM structure of the microtubule bound to GAS2-GAR, the authors should show the electron maps of alpha and beta tubulin differences used for subunit assignment.
8. The authors should also compare the binding of other microtubule-binding domains with GAS2-GAR, which can provide insights into GAR domain biochemical similarities to other microtubule-binding domains.
9. The manuscript does not explain how the polarity of microtubules was assigned in Figure 5.
10. The effects of GAR on microtubule dynamics are very interesting. However, the current data in the manuscript makes it hard to interpret or appreciate the differences. It would be ideal to show a movie and add individual data points as scatter/violin plots in Figure 6H.
11. It is also unclear why GAR affects microtubule dynamics or whether it binds to free tubulin, and more discussion is needed.
12. The manuscript also highlights the spectraplakins family members. A recent study extensively characterized MACF1 and showed microtubule ring structures (Jijumon et al., Nature Cell Biology 2022). The authors also observed similar ring-like structures, as shown in Figure 6, so discussing these results with other observations would enhance the paper.

Referee #2:

In their paper "Dimerization [sic] GAS2 mediates microtubule and F-actin crosslinking", An and colleagues present a comprehensive structural and biochemical dissection of actin-microtubule mediated crosstalk by the spectraplakins family member GAS2. They present cryo-EM structures of both the actin-binding (CH3) and microtubule binding (GAR) domains

engaged with their respective filaments, identifying key residues mediating binding which are validated with thorough biochemical assays. They furthermore show that the C-terminal segment of the protein mediates dimerization, which is important for the formation of higher-order microtubule-F-actin bundle assemblies. Notably, this segment is cleaved during apoptosis, providing mechanistic insights into how cytoskeletal rearrangements during apoptosis are coordinated at the molecular scale. Finally, they provide evidence that the GAR domain impacts microtubule nucleation and dynamics, although the physiological importance of this activity is less clear.

Overall, I found this to be a very thorough and insightful characterization of a microtubule-actin crosstalk factor, an area of cell biology which is still largely mysterious. The point mutants identified will be very useful for future cellular and in vivo characterization of the protein using separation-of-function mutations, an approach which will be important for making headway in the cytoskeletal crosstalk field. Broadly speaking, the array of techniques employed is impressive and insightful, and I believe the major conclusions of the study are both technically and conceptually sound. The study thus represents a significant advance.

My major critique focuses on the cryo-EM structural analysis, which I believe can be addressed through revision.

Major concern:

The authors claim they have obtained a 2.8 Angstrom CH3-F-actin reconstruction, and a 3.2 Angstrom GAR-MT reconstruction. However, these resolution assessments are driven by the filaments themselves, which feature 100 % occupancy and a high degree of order.

Examining local resolution plots (Fig. S3F, S2A), as well as visual inspection of the maps and models providing by the authors, shows that the resolution of the bound factors, and their interface with the filaments, is substantially worse. Consistently, both maps feature a great deal of noise (which it seems the authors hid using the "hide dust" function in Chimera or the equivalent in their figures).

This issue is particularly exacerbated for the GAR domain. Beyond the direct interface with the microtubule, the GAR domain density is discontinuous, and even large bulky side chains are not in the density. This is furthermore reflected in the validation reports, where the atomic models for the binding domains (particularly the GAR domain, but also the CH3 domain) feature poor atomic inclusion in the map, low Q scores (representing map-model overlap in sidechain regions), and poor geometry statistics relative to other cryo-EM structures at similar resolution in the PDB.

On the bright side, inspection the maps shows that the interfaces between the domains and the filaments are well-resolved. This is further supported by the mutagenesis / binding assays. Since resolving the interfaces is the major focus on the structural analysis, I believe the authors conclusions are largely OK. However, I believe the current presentation is misleading, and the sub-optimal structures could be improved through additional computational analysis.

Suggestions for improving analysis: In addition to flexibility at the domain-filament interfaces, the most likely culprit for poor domain density in both cases is low occupancy / substoichiometric binding. I encourage the authors to pursue a symmetry expansion approach, as has been done for analysis of troponin (PMID 31919429), as well as the T-plastin study cited in the manuscript (PMID: 36067297). This would allow them to perform focused 3D classification on the binding interface (using a mask around the domain) without alignment, considering each potential binding interface independently.

Whether or not the density can be improved, the authors should also consider using Isolde for interactive MDFF fitting. This software supports placing adjustable restraints on the starting input model, as well as fitting multiple maps simultaneously (e.g. a low-pass filtered map for the region of the domains not contacting the filaments, as well as the current maps of the interfaces).

Whether or not these approaches are successful, I believe the authors should give a more balanced presentation of the structural data. This includes:

-Showing images of the model-map overlap in both the good and bad regions of the binding domains in the Figures.

-Discussion of the implications of the local resolution gradients in the main text, when the structures are introduced.

-Tempering claims about parts of the domains that are not well-resolved (e.g. the Zn²⁺ binding moiety of the GAR domain: their map definitely cannot resolve a Zn²⁺ ion).

Minor Concerns:

-In the introduction, the authors refer to the "microfilament system". I suggest the term "actin cytoskeleton" would be more accessible to a general audience (or a clarification if something else is meant).

-The physiological implications of GAS2's effects on MT nucleation / polymerization dynamics are unclear. Is there prior evidence of GAS2 impacting MT dynamics in cells? Are the concentrations where these effects were observed relevant in a cell?

- In the methods, the authors say "Images were collected with a Gatan-LS Energy Filter". It would be useful to explicitly state what the detector is in this system.
- In the methods, the authors refer to selecting a subset of data with "a maximum metadata value of 5.5". What sort of metadata are they referring to?
- The custom code for analyzing MT polarity should be made publicly available as a condition for publication.
- Fig. S4: The pLDDT plot is difficult to understand. It would be more useful to show this mapped on the structure.
- Filament polarity should be indicated on at least one panel in each figure where structures are shown.
- Title: "Dimerization of GAS2 mediates microtubule and F-actin crosslinking"

Referee #3:

GAS2 plays a crucial role in crosslinking microtubules (MT) and F-actin through its CH and GAR domains, which are essential for organizing the cytoskeleton in cochlear supporting cells. The manuscript authored by An and coauthors studied the structural underpinnings of GAS2's interaction with F-actin and MT using cryo-electron microscopy. They further validated the structure models through mutation analysis. They suggested that GAS2 can promote microtubule nucleation and rescue and that the C-terminal region is important for its function, likely through dimerization. Overall, most of the results of this study are convincing, and the structural information is valuable.

The major concern of this review is their conclusions related GAS2's active towards MT nucleation and polymerization, as well as the dimerization function of the C-terminal region.

1. Figure 6B and 6E, why can we see MT formation in Figure 6B with GAS2 GAR domain but not in Figure 6E with GAS2 FL? Shouldn't the full-length protein be more potent?
2. Figure 6A and 6B, can the process of MT formation be monitored by TIRF microscopy assay?
3. Figure 6H and 6I, Figure S5E-G, it appears that the GAS2 GAR domain primarily accumulate on MT seeds rather than dynamic MTs, and the binding appears to be weak. How does this impact MT dynamics? It's very difficult to understand how it works, especially the very slow MT growth. The author should also utilize the full-length protein to conduct the same assay. The significance of the critical residues identified by the pelleting assay can also be examined in a TIRF microscopy assay.
4. The author can use TIRF microscopy assay to compare the MT binding activity of GAS2 FL and deletion mutants towards single, parallel, and anti-parallel MTs, which would strengthen the conclusion that GAS2 can bundle both parallel and anti-parallel MTs.
5. Figure S4B and C, the predicted model of the dimerized C-terminal region should be experimentally verified by mutation analysis by gel filtration and functional assay.

Minor points,

6. Figure 2, the author should explain why mutating some residues has stronger effects than others.
7. The authors provide the Kd for the GAS2 CH domain in Figure 2K. What are the Kd values for GAS2 FL and GAS2 delta-C in Figure 1C? The author should also compare them.
8. Figure 7, can the author use TIRF microscopy assay to monitor the crosslinking of MT and F-actin by GAS2?

Dear Editor,

Thank you so much for giving us an opportunity to revise our manuscript. We sincerely appreciate the editor and reviewers for their constructive comments and suggestions to our manuscript entitled “Dimerization of GAS2 mediates microtubule and F-actin crosslinking” (Manuscript EMBOJ-2024-119009).

Those comments are all valuable and very helpful for revising and improving our paper, as well as the important guiding significance to our following research. We have presented detailed responses to the reviewer’s comments. We have carefully revised the original manuscript according to the reviewers' comments. All revised comments are marked in the revised manuscript.

The comments and responses are listed on the following pages.

Referee #1:

This manuscript, "Dimerization GAS2 mediates microtubule and F-actin crosslinking," by An et al., reports the structure of GAS2 domains bound to actin and microtubule filaments. The structure-function work is of very good quality, and the manuscript attempts to present a model of GAS2-mediated crosslinking using various structural and biochemical methods. Few studies have reported F-actin and microtubule crosslinkers' structural and biochemical properties. Therefore, this manuscript will be of broad interest to the cytoskeleton and cell biology community. I have a few suggestions and recommendations for improving the manuscript before publication.

1. The authors have described the hearing loss mutations and correlated them with their biochemical work in the discussion and the model figure. It will be impactful if they include a sentence about this in their abstract.

Thank you for your insightful suggestion. We have incorporated a sentence into the abstract emphasizing the connection between hearing loss mutations and our biochemical and structural findings. This addition underscores the clinical relevance of our study. The revised abstract now includes the sentence as follows (lines 34-36 with a yellow highlight background in the main text): "Our structure and biochemistry study provide insights into truncated GAS2 mutants which resulted in hearing loss, offering a better understanding of GAS2's role in auditory dysfunction."

2. In Figure 1A and Appendix Figure S1, the phosphorylation and caspase cleavage sites and the inherited hearing loss mutations can be marked.

Thank you for the suggestion. We have marked the caspase cleavage sites in Figure 1A first panel with a scissors icon and have added this information to the figure legend with yellow highlight in line 889 as the following statement: "The scissors icon indicates the caspase-3 cleavage site at D279."

We added the inherited hearing loss mutation in Figure 1A bottom panel and have added this information to the figure legend with yellow highlight in lines 894-896 as the following statement: "The sixth panel denotes the mutation (c.723+1G>A) associated with hearing loss, which leads to intron retention and results in a C-terminally truncated GAS2 protein originating from the GAR domain."

Figure 1A

A

Regarding phosphorylation sites: Since the phosphorylation pattern of GAS2 is not uniform and numerous amino acids can be phosphorylated during the cell cycle, we decided not to include these sites in the figure to avoid potential misrepresentation.

We present a schematic of the MACF1 protein in Appendix Figure S1A. We did not change this figure because there is no evidence that caspase cleaves MACF1.

3. The GAS2-CH3 domain boundaries used for cryoEM work need to be mentioned in the methods and the beginning of results describing cryoEM work.

Thank you for your suggestion regarding the inclusion of the GAS2-CH3 domain boundaries. We have added the description of the GAS2-CH3 domain boundaries in the **Methods** section (line 576) and in the **Results** section describing the CryoEM work (lines 157-158). These additions are highlighted in yellow.

Methods part:

“For F-actin-GAS2-CH3 complex, 40 μ L of Actin (4 μ M polymerized in F-buffer) was mixed with 20 μ L of 56 μ M GAS2-CH3 at room temperature for 40 minutes in F-buffer, followed by the addition of another 20 μ L of 56 μ M GAS2-CH3 (residues 1-201) for further incubation for 40 minutes.”

Results part:

“We used GAS2-CH3 (residues 1-201, Fig 1A fifth panel), the minimum construct that can bind to F-actin, and obtained a high-quality density map with a local resolution gradient better than 3.0 \AA , as judged by the FSC curve (Table 1, Fig 2A, Figure EV2A).”

4. In Figure 2 and Appendix Figure S2, the polarity of the F-actin should be marked.

Thank you for your suggestion regarding the indication of F-actin polarity. We have added clear markings of the F-actin polarity in **Figure 2** and **Figure EV2** (Appendix Figure S2) to improve clarity and aid interpretation.

Figure 2 A/C

Figure EV2B

We appreciate your valuable feedback, which has helped enhance the quality of our figures.

5. Similarly, in Appendix Figure S2 B, the electron maps of critical interactions, as shown in Figure 2 D, can be shown instead.

Thank you for the suggestion. We have updated **Figure EV2B** (Appendix Figure S2B) to include the electron density maps highlighting the critical interactions similar to those shown in **Figure 2D**. This figure provides a clearer view of the structural details and enhances the consistency between the main text and supplementary figures.

Figure EV2B

6. The structure-function analysis of GAS2-CH3 binding to F-actin shows that ABD1 seems dispensable, similar to the UTRN studies by Kumari et al. However, the authors have mentioned that the hydrophobic pocket of ABD1 is critical. This needs some reconciliation between their results and conclusions.

We thank the reviewer for highlighting the need to reconcile our results with the findings of Kumari et al. (2020). Our biochemical study demonstrated that GAS2-CH3 Δ N, which lacks the ABD1 region, significantly reduces its F-actin binding affinity (Fig. 2K). This suggests that ABD1 is critical for GAS2-CH3's strong interaction with F-actin in vitro. In contrast, the

study by Kumari et al. (2020) showed that the CH1 domain of utrophin, containing only ABD2' and ABD2, is sufficient for F-actin interaction in a cellular context. This discrepancy may arise from the distinct experimental systems: biochemical assays typically measure direct, high-affinity interactions in a simplified environment, while cellular studies capture binding under complex, crowded conditions influenced by additional factors such as accessory proteins, post-translational modifications, or cytoplasmic constraints. Our future research direction and studies combining biochemical and cell biology approaches elucidate the role of GAS2-CH3 ABD1 in different contexts.

We have addressed this point in the revised manuscript by adding the following clarification to the discussion section **lines 351-356 with yellow highlight**:

“Our biochemical study demonstrated that GAS2-CH3 Δ N, which lacks the ABD1 region, significantly reduces its F-actin binding affinity (Fig 2K). In contrast, cellular co-localization analysis by Kumari et al. (2020) showed that the CH1 domain of utrophin, containing only ABD2' and ABD2, is sufficient for F-actin interaction in cells (Kumari et al, 2020). These findings suggest that ABD1 is crucial for GAS2-CH3's F-actin binding in vitro, but its necessity in cellular contexts requires further investigation.”

We appreciate your insightful comment and believe this revision better interprets the ABD1 domain's role.

7. In the cryoEM structure of the microtubule bound to GAS2-GAR, the authors should show the electron maps of alpha and beta tubulin differences used for subunit assignment.

Thank you for your suggestion to show the electron maps highlighting the differences between α - and β -tubulin in our cryo-EM structure. We have added a supplementary figure (**Fig EV3F**, also as the following) that provides a zoomed-in view of the cryo-EM reconstruction from the lumen. The red dashed circles highlight the density regions corresponding to the S9-S10 loop, which has different lengths in α - and β -tubulin (α -tubulin has a longer S9-10 loop, **lines 230-231 with yellow highlight**), aiding in subunit assignment.

Here is the statement in the main text:

“ α - and β -tubulin are similar, but we distinguished them based on the difference in the length

of the S9-S10 loop (Fig. EV3F).”

We appreciate your feedback, which has helped us present this structural information more clearly.

8. The authors should also compare the binding of other microtubule-binding domains with GAS2-GAR, which can provide insights into GAR domain biochemical similarities to other microtubule-binding domains.

Thank you for your suggestion to compare the GAS2-GAR domain with other microtubule-binding domains. We have added a new discussion section (lines 388-393 labeled with yellow highlight) highlighting the similarities between the GAS2-GAR domain and the microtubule-binding domains of kinesin and dynein as follows:

“To identify the GAS2-GAR domain binding similarity that may be involved in the interaction of tubulin with these motor proteins, we used the footprint method (Liu et al, 2017) to inspect potential contacts between tubulin and the binding partner. By comparison footprint on MTs, we observed that the GAS2-GAR domain, kinesin (PDB: 6WWI), and dynein (PDB: 6RZA) share a similar interaction surface on $\alpha\beta$ -tubulin dimer, involving H11, H12 of α -tubulin and H4, H5, H12 of β -tubulin (Figure EV5I).”

We believe this comparison provides valuable insights into the biochemical similarities of the GAR domain and enhances the understanding of its interaction with microtubules.

9. The manuscript does not explain how the polarity of microtubules was assigned in Figure 5.

Thank you for your comment regarding the assignment of microtubule polarity in Figure 5. We have clarified this process in the Materials and Methods section under the "Microtubule Polarity Trace" subsection (lines 687-693) as follows.

“The polarity of individual microtubules was determined by matching the 3D structure reconstruction of the GAS2-GAR-MT complex to the image. First, the polarity of the final GAS2-GAR-MT was established by using ChimeraX based on the known microtubule structure. Then, we assign the polarity to individual particles in the 2D raw micrographs based on the tilt and ψ angle data stored in the RELION Star file. This polarity information was then used to overlay arrows onto the raw micrographs, where the arrow orientation reflects the polarity from the microtubule minus end to the plus end. Refer to the Code EV file for detailed script information.”

10. The effects of GAR on microtubule dynamics are very interesting. However, the current data in the manuscript makes it hard to interpret or appreciate the differences. It would be ideal to show a movie and add individual data points as scatter/violin plots in Figure 6H.

Thank you for your insightful suggestion regarding the presentation of GAR's effects on microtubule dynamics. We have added the supplementary movie (**Movie EV6-8**) to illustrate these dynamics more clearly and have updated **Figure 6H** to display individual data points using scatter and violin plots. This revision enhances the clarity and interpretability of the differences observed.

Figure 6H/I

We appreciate your valuable feedback, which has significantly improved the presentation of our data.

11. It is also unclear why GAR affects microtubule dynamics or whether it binds to free tubulin, and more discussion is needed.

Thank you for this insightful comment. We have expanded our discussion to address these points as follows:

Microtubule Dynamics (lines 416-425 with yellow highlight):

" Our microtubule dynamics assay demonstrated that the GAS2-GAR domain stabilizes MTs by inducing a pausing state following polymerization. Interestingly, the GAR domain binds to regions on the MT lattice that overlap with those of motor proteins (Figure EV5I). This finding suggests that the GAR domain may play a regulatory role akin to that of motor proteins and other microtubule-associated proteins. Zhang et al. (2018) reported that GMPCPP-stabilized MTs with 14 protofilaments exhibit a dimer rise of approximately 83.96 Å. Kinesin-1, known to stabilize MTs, reduces this spacing by 0.7 Å (Zhang et al, 2018). While in our study, we observed that GAS2-GAR-decorated MTs displayed a dimer rise of 82.25 Å, indicating a 1.7 Å reduction. This lattice compaction induced by GAS2-GAR likely contributes to the enhanced MT stability and the observed pausing of MT dynamics. Nevertheless, the influence of GAS2 on microtubule dynamics under cellular conditions remains elusive, and further

investigations are needed to uncover the precise mechanisms of how the GAS2 protein regulates microtubule dynamics."

Binding to Free Tubulin (lines 399-402 with yellow highlight):

" However, we did not directly observe the incorporation of tubulin rings into sheets or microtubules, and we currently lack direct evidence for GAS2-GAR binding to free tubulin; such interactions could promote nucleation or affect the pool of polymerizable tubulin subunits; future studies should clarify this."

12. The manuscript also highlights the spectraplakin family members. A recent study extensively characterized MACF1 and showed microtubule ring structures (Jijumon et al., Nature Cell Biology 2022). The authors also observed similar ring-like structures, as shown in Figure 6, so discussing these results with other observations would enhance the paper.

Thank you for pointing out the relevance of the recent study on MACF1 by Jijumon et al. (Nature Cell Biology, 2022) and suggesting a comparative discussion. We have reviewed their findings and repeated similar experiments.

However, there are significant differences between their observations and ours:

- 1. Structural Differences:** The ring-like structures reported for MACF1 have a diameter of approximately 1 μm , while the rings we observed in our negative staining experiments are much smaller, with a diameter of about 50 nm. This size difference suggests a distinct mechanism or function for the GAS2-GAR-induced rings compared to MACF1.
- 2. Experimental Observations:** Due to the smaller size of our rings, they were not detectable by TIRF microscopy. Instead, we observed a gradual transition from a spot-like structure to microtubule growth, as demonstrated in the **Movie EV9**. This dynamic behavior contrasts with the large, stable rings reported for MACF1.

We have expanded the discussion in the revised manuscript to highlight these differences and to propose that while both proteins promote microtubule ring formation, they may do so through different mechanisms or structural interactions. This comparison underscores the diversity within the spectraplakin family and the unique properties of GAS2-GAR. The discussion is as follows **(lines 402-411 with yellow background):**

“Recent studies, such as Jijumon et al. (Jijumon et al, 2022), have shown that MACF1, a member of the spectraplakin family, promotes the formation of large tubulin ring structures with diameters of approximately 1 μm . These rings appear to serve as a template for microtubule nucleation, stabilizing tubulin dimers and facilitating microtubule growth. We observed a distinct phenomenon: GAS2-GAR induces the formation of much smaller ring-like structures (~50 nm in diameter). Unlike MACF1, these structures were not visible using TIRF microscopy, but we detected a gradual transition from spot-like aggregates to microtubule growth in the presence of GAS2-GAR, as shown in Movie EV9. This suggests that the mechanism by which GAS2-GAR influences microtubule ring structure may differ significantly from that of MACF1.”

Referee #2:

In their paper "Dimerization [sic] GAS2 mediates microtubule and F-actin crosslinking", An and colleagues present a comprehensive structural and biochemical dissection of actin-microtubule mediated crosstalk by the spectraplakins family member GAS2. They present cryo-EM structures of both the actin-binding (CH3) and microtubule binding (GAR) domains engaged with their respective filaments, identifying key residues mediating binding which are validated with thorough biochemical assays. They furthermore show that the C-terminal segment of the protein mediates dimerization, which is important for the formation of higher-order microtubule-F-actin bundle assemblies. Notably, this segment is cleaved during apoptosis, providing mechanistic insights into how cytoskeletal rearrangements during apoptosis are coordinated at the molecular scale. Finally, they provide evidence that the GAR domain impacts microtubule nucleation and dynamics, although the physiological importance of this activity is less clear.

Overall, I found this to be a very thorough and insightful characterization of a microtubule-actin crosstalk factor, an area of cell biology which is still largely mysterious. The point mutants identified will be very useful for future cellular and in vivo characterization of the protein using separation-of-function mutations, an approach which will be important for making headway in the cytoskeletal crosstalk field. Broadly speaking, the array of techniques employed is impressive and insightful, and I believe the major conclusions of the study are both technically and conceptually sound. The study thus represents a significant advance.

My major critique focuses on the cryo-EM structural analysis, which I believe can be addressed through revision.

Major concern:

The authors claim they have obtained a 2.8 Angstrom CH3-F-actin reconstruction and a 3.2 Angstrom GAR-MT reconstruction. However, these resolution assessments are driven by the filaments themselves, which feature 100 % occupancy and a high degree of order.

Examining local resolution plots (Fig. S3F, S2A), as well as visual inspection of the maps and models provided by the authors, shows that the resolution of the bound factors, and their interface with the filaments, is substantially worse. Consistently, both maps feature a great deal of noise (which it seems the authors hid using the "hide dust" function in Chimera or the equivalent in their figures).

This issue is particularly exacerbated for the GAR domain. Beyond the direct interface with the microtubule, the GAR domain density is discontinuous, and even large bulky side chains are not in the density. This is furthermore reflected in the validation reports, where the atomic models for the binding domains (particularly the GAR domain, but also the CH3 domain) feature poor atomic inclusion in the map, low Q scores (representing map-model overlap in

sidechain regions), and poor geometry statistics relative to other cryo-EM structures at similar resolution in the PDB.

On the bright side, inspection of the maps shows that the interfaces between the domains and the filaments are well-resolved. This is further supported by the mutagenesis / binding assays. Since resolving the interfaces is the major focus on the structural analysis, I believe the authors' conclusions are largely OK. However, I believe the current presentation is misleading, and the sub-optimal structures could be improved through additional computational analysis.

Suggestions for improving analysis: In addition to flexibility at the domain-filament interfaces, the most likely culprit for poor domain density in both cases is low occupancy / substoichiometric binding. I encourage the authors to pursue a symmetry expansion approach, as has been done for the analysis of troponin (PMID 31919429), as well as the T-plastin study cited in the manuscript (PMID: 36067297). This would allow them to perform focused 3D classification on the binding interface (using a mask around the domain) without alignment, considering each potential binding interface independently.

Thank you for your valuable suggestion regarding symmetrical expansion and focused 3D classification.

GAS-GAR-MT complex

We implemented these approaches for the GAS2-GAR-tubulin complex structure using CryoSPARC to enhance the density quality.

Specifically:

1. We picked more particles utilizing automated template picking with a good template. It resulted in 4 times more particles (44,922 vs 189,762).
2. The protofilament density was symmetry expanded and subtracted. Then, the structure was refined by focusing two tubulin dimers with 1 GAR mask, increasing the overall resolution to 2.8 Å.

Local resolution analysis for MiRP results map indicates that the GAR domain region initially showed lower resolution (most parts of GAR density regions are worse than 3.5 Å resolution, and some regions are worse than 4.5 Å resolution). After CryoSPARC local refinement, GAR density showed significant improvement, 3.5-4.0 Å resolution, as highlighted in **Appendix Figure S1A**, and the following figure shows the local map resolution at the same color key for MiRP and CryoSPARC density map. Further comparison of the MiRP and CryoSPARC maps revealed that the CryoSPARC map displayed improved continuity and clarity in the density map.

However, the distinction between alpha and beta tubulin became less pronounced after CryoSPARC local refinement, and the GAS2-GAR density appeared within both intra- and inter-tubulin dimer regions (shown in the following figure). So, we have used MiRP density map for scientific display and used CryoSPARC density map to refine the PDB structure by Phenix and ISOLDE for further.

We also added the CryoSPARC data analysis processing for GAS2-GAR-MT in the method part (line 634-657 labeled with green highlight) as follows:

“For high-resolution analysis of the MT-GAS2-GAR complex, the images were reprocessed using CryoSPARC (version 4.62) (Punjani et al, 2017). Imported movies were first processed using patch motion correction and patch CTF estimation. Filaments were manually picked and segmented into 540×540 pixel boxes at an 82-angstrom step, generating approximately 1,000 particles. Particles were 2D classified, good particles were selected and used with the Filament Tracer tool to pick additional filaments. The picked filaments were segmented at an 82-angstrom pitch, binned fourfold into 200×200 pixel boxes, and subjected to 2D classification to remove non-microtubule particles. After several rounds of 2D classification for cleanup, the particles were subjected to heterogeneous refinement using an 11-16 protofilament microtubule reference, also used for MiRP2 analysis, for further classification. Following

several rounds of classification, the major homogeneous particles corresponding to 14-protofilament microtubules were selected and re-extracted at 1.0375 Å/pixel in 400 × 400 pixel boxes. Particles were refined using Helical Refinement without symmetry, resulting in a 3.7 Å resolution map. Subsequently, symmetry expansion was applied with a helical twist of -25.75° and a helical rise of 8.88 Å, based on the MiRP2 analysis of the 14-protofilament MT-GAS2-GAR. After symmetry expansion, particles were subtracted using a mask corresponding to two tubulin heterodimers with one GAS2-GAR domain generated from the MiRP2-derived model and a soft mask created with Relion 5.0. The subtracted particles were then refined using Local Refinement with non-uniform refinement and the same focus mask used for particle subtraction, resulting in the final map. Before final refinement, we conducted 3D classification; however, the classified classes did not show significant differences, so this strategy was not adopted. The global resolution of the composite maps was estimated using gold-standard Fourier shell correlation (FSC = 0.143), and the local resolution was determined using the local resolution estimation tool of CryoSPARC. Image processing and map quality are summarized in Appendix Figure S1A."

GAS2-CH3-F-actin

We also explored these approaches with the GAS2-CH3 domain to improve the density map quality. Specifically:

1. Symmetry Expansion Approach:

We attempted symmetry expansion on our dataset to enhance particle count and resolution. However, given our helical parameters—an extraction interval of 27.4 Å (equal to the helical rise) and C1 symmetry—the calculated symmetry expansion factor n is 1. Consequently, no particle multiplication occurred, rendering this approach ineffective in our case.

2. Focused 3D classification:

Further 3D classification presents another opportunity to improve the density of the binding protein. In a 3D classification using helical symmetry averaging (T=4, N=4) from the final dataset, one cluster accounted for almost all particles (97.4%), indicating that further selection was ineffective. Subsequently, we performed another 3D classification using a mask that included one binding protein along with its interface region on the actin subunits (as in the following figure) without helical symmetry. A class with a high-density binding protein was selected, and 3D refinement without helical averaging was carried out. However, in this case, the local resolution of the actin-binding protein density was substantially worse, which could be attributed to the mask being too small. Thus, we can claim the current map presented in the manuscript appears to be the best representation based on the available data.

Whether or not the density can be improved, the authors should also consider using Isolde for interactive MDFF fitting. This software supports placing adjustable restraints on the starting input model, as well as fitting multiple maps simultaneously (e.g., a low-pass filtered map for the region of the domains not contacting the filaments, as well as the current maps of the interfaces).

Thank you for your suggestion regarding the use of ISOLDE for interactive Molecular Dynamics Flexible Fitting (MDFF). We have implemented this approach in our study and have used ISOLDE to refine the structures of both GAS2-CH3-F-actin and GAS2-GAR-MT. Specifically, we utilized ISOLDE to fit our model to the cryo-EM maps, including both the current maps of the binding interfaces and low-pass filtered maps for regions of the domains not in direct contact with the filaments. This allowed us to apply adjustable restraints to the starting model, improving the accuracy of our fitting process. The use of ISOLDE provided valuable insights into the refinement of the structure and helped to enhance the consistency between the model and the experimental data. After we revised the structure, we deposited the structure to PDB and EMDB again. The validation report of GAS2-CH3-F-actin complex has greatly improved.

For GAS2-GAR-MT complex also has greatly improvement. Combining our biochemistry experiments and mutation experiments validated the main binding sites between GAR and MT. There is a minor revision in the main text regarding GAR's interaction with MT (lines 246-248 with green background):

“The density map and atomic models reveal GAR domain α -helix2 K266 close to β -tubulin E431 and H267 proximity to β -tubulin E196, suggesting electrostatic interactions (Fig. 4H).”

We believe these approaches have significantly contributed to the quality of the final structure. Here is the PDB deposition ID and password. If necessary, please check them.

GAS2-CH3-F-actin PDB deposition ID link: D_1300051871, **Password:** Unavailable

GAS2-GAR-MT PDB deposition ID link: D_1300051864, **password:** Unavailable

We appreciate the reviewer's suggestion, and we have made sure to include ISOLDE-based refinements as part of our methodology.

Whether or not these approaches are successful, I believe the authors should give a more balanced presentation of the structural data. This includes:

-Showing images of the model-map overlap in both the good and bad regions of the binding domains in the Figures.

Thank you for the suggestion. We have incorporated images of the model-map overlap for both the well-resolved ("good") and poorly-resolved ("bad") regions of the binding domains in the revised Figures. For GAS2-CH3-F-actin, good regions were shown in **Figure EV2C** and bad regions were shown in **Figure EV2D**.

Figure EV2C/D

For GAS2-GAR-tubulin, model-map overlaps are shown in **Appendix Figure S1E**. These images provide a clear visualization of the density quality across different regions and illustrate the model's fit to the map. We believe this addition enhances the clarity and completeness of the structural analysis presented in the manuscript.

Appendix Figure S1E

-Discussion of the implications of the local resolution gradients in the main text, when the structures are introduced.

Thank you for your valuable comment regarding the discussion of local resolution gradients. We have added a detailed explanation in the main text to highlight the implications of these gradients, particularly their impact on structural interpretation and biological function.

1. For GAS2-CH3-F-actin complex (lines 158-160, and lines 163-167 labeled with green highlight), we state:

" We used GAS2-CH3 (residues 1-201), the minimum construct that can bind to F-actin and obtained a high-quality density map with a local resolution gradient better than 3.0 Å, as judged by the FSC curve (Table 1, Fig. 2A, Fig. EV2A). "

"The high-resolution regions of the GAS2-CH3 domain, located at the interface with F-actin, enabled accurate modeling and elucidation of their interactions (Fig. EV2B, C). Conversely, the lower-resolution regions corresponding to the loops of GAS2-CH3, situated distal to the F-actin interface, exhibited poor map fitting, indicating their intrinsic flexibility (Fig. EV2D). "

2. We discussed the local resolution gradients for the GAS2-GAR MT structure in the main text (lines 217-227, and lines 231-235 labeled with green background). Specifically, we described how the high-resolution regions enabled precise modeling of the microtubule-binding interface, while the lower-resolution regions indicate flexible domains likely important for dynamic interactions, as follows:

"Our analysis achieved a 3.2 Å resolution of 14-pf microtubules complexed with the GAS2-GAR domain (Fig. EV3C). Local resolution estimates varied from 2.5 to 3.5 Å for tubulins and 3.5 to 4.5 Å for GAR domain molecules (Fig. EV3E). The density map reveals that the GAR domain is inserted into the tubulin intradimer groove (Fig. 4A, B). Notably, the resolutions of N- and C-termini and certain loops of the GAS2-GAR domain dip below 4 Å, possibly due to its flexibility (Fig. EV3E).

For a higher-resolution analysis of the GAS2-GAR-MT complex, we reprocessed the cryo-EM images using symmetry expansion and achieved a 2.8 Å overall resolution. This improved map enabled us to build an atomic model and observe amino acid sidechains at the GAR-MT interface (Appendix Fig. S1A). However, the distinction between α - and β -tubulin became less pronounced (Appendix Fig. S1B)."

" Manual modeling and iterative refinement resulted in the final atomic model of the GAS2-GAR-MT complex (Fig. 4C, Fig. EV3G, Appendix Fig. S1D). The MT-binding

surface of GAR was modeled well to fit into the density map (Appendix Fig. S1E), while the flexible N- and C-termini and some loops were outside the map. "

We appreciate your insightful feedback, which has allowed us to enhance the scientific rigor in our structural description.

-Tempering claims about parts of the domains that are not well-resolved (e.g., the Zn²⁺ binding moiety of the GAR domain: their map definitely cannot resolve a Zn²⁺ ion).

Thank you for your valuable comment regarding the Zn²⁺ binding moiety of the GAR domain. We agree that our map does not provide sufficient resolution to confidently model a Zn²⁺ ion. Therefore, we have removed any mention of Zn²⁺ binding from the manuscript to avoid overinterpretation.

We appreciate your insight, which has helped us ensure our claims are fully supported by the data.

Minor Concerns:

-In the introduction, the authors refer to the "microfilament system". I suggest the term "actin cytoskeleton" would be more accessible to a general audience (or a clarification if something else is meant).

Thank you for your helpful suggestion regarding terminology. We agree that "actin cytoskeleton" is more accessible to a general audience. Therefore, we have replaced "microfilament system" with "actin cytoskeleton" throughout the manuscript for clarity (lines 84 and 86, labeled with green background). We appreciate your feedback on improving the accessibility of our work.

-The physiological implications of GAS2's effects on MT nucleation / polymerization dynamics are unclear. Is there prior evidence of GAS2 impacting MT dynamics in cells? Are the concentrations where these effects were observed relevant in a cell?

Thank you for raising this important point regarding the physiological implications of GAS2's effects on microtubule (MT) dynamics. At present, there is no direct evidence showing that GAS2 impacts MT nucleation or polymerization dynamics in cells. The observed effects in our in vitro experiments highlight a potential role, but further investigation is needed to confirm their relevance in a cellular context, and this is our future research direction.

We acknowledge this limitation in the revised manuscript (lines 425-427 with green highlight) as follows:

" Nevertheless, the influence of GAS2 on microtubule dynamics under cellular conditions remains elusive, and further investigations are needed to uncover the precise mechanisms of how the GAS2 protein regulates microtubule dynamics."

We appreciate your insightful feedback and hope this revision addresses your concerns.

-In the methods, the authors say "Images were collected with a Gatan-LS Energy Filter". It would be useful to explicitly state what the detector is in this system.

Thank you for your suggestion to specify the detector used in our imaging system. We have revised the Methods section to include this information. The updated text now reads (line 589 with a green highlight):

"Images were collected with a Gatan-LS Energy Filter (Gatan, Pleasanton, CA) with a slit width of 20 eV and a Gatan K3 Summit direct electron detector in the electron counting mode."

We appreciate your feedback, which has helped us improve the clarity and completeness of our methods.

-In the methods, the authors refer to selecting a subset of data with "a maximum metadata value of 5.5". What sort of metadata are they referring to?

Thank you for your question regarding the metadata mentioned in our Methods section. We have clarified this in the revised manuscript. The sentence now specifies that the "maximum resolution cutoff of 5.5 Å" refers to the resolution determined from the CTF (contrast transfer function) estimation of the raw micrographs (lines 603-605 with green highlight). We appreciate your attention to detail and believe this revision clarifies the methodology.

The revised sentence is as follows:

"For F-actin-GAS2-CH3 complex, 5,756 micrographs were selected by subset selection in RELION (Scheres, 2012) with a maximum resolution cutoff of 5.5 Å determined from the CTF resolution estimation of the raw micrographs. "

-The custom code for analyzing MT polarity should be made publicly available as a condition for publication.

Thank you for highlighting the importance of making the custom code publicly available to analyze microtubule (MT) polarity. To ensure transparency and reproducibility, we have

included the code as **supplementary material under Code EV** in the revised manuscript. This will allow other researchers to access and utilize the code for their analysis. We appreciate your suggestion and have ensured that all necessary documentation is provided alongside the code for ease of use.

-Fig. S4: The pLDDT plot is difficult to understand. It would be more useful to show this mapped on the structure.

Thank you for your helpful comment regarding the pLDDT plot in Fig. S4. Combined with reviewer3's comments, we have moved this figure to the Discussion section as **Appendix Figure S5 A** and updated it to show the pLDDT plot mapped directly onto the structure for better clarity and understanding. We appreciate your suggestion, which has improved the presentation of this data.

Appendix Figure S5 A

-Filament polarity should be indicated on at least one panel in each figure where structures are shown.

Thank you for your comment regarding the indication of filament polarity. We have ensured that F-actin and microtubule (MT) polarity are indicated in all figures (**Fig 2A, C, K; Fig 3A; Fig EV2B; Fig EV 3E, F, G**). This information has been added to the appropriate panels to provide clarity.

-Title: "Dimerization of GAS2 mediates microtubule and F-actin crosslinking"

Thank you for your suggestion regarding the title. We have updated it to "Dimerization of GAS2 mediates microtubule and F-actin crosslinking," as recommended.

Referee #3:

GAS2 plays a crucial role in crosslinking microtubules (MT) and F-actin through its CH and GAR domains, which are essential for organizing the cytoskeleton in cochlear supporting cells. The manuscript authored by An and coauthors studied the structural underpinnings of GAS2's interaction with F-actin and MT using cryo-electron microscopy. They further validated the structure models through mutation analysis. They suggested that GAS2 can promote microtubule nucleation and rescue and that the C-terminal region is important for its function, likely through dimerization. Overall, most of the results of this study are convincing, and the structural information is valuable.

1. Figure 6B and 6E, why can we see MT formation in Figure 6B with GAS2 GAR domain but not in Figure 6E with GAS2 FL? Shouldn't the full-length protein be more potent?

We appreciate the reviewer's thoughtful question. We hypothesize that the full-length GAS2 protein contains the CH3 domain, which may inhibit the GAR domain's ability to bind to MTs. In the absence of the CH3 domain, the GAR domain is released from this inhibition, allowing it to exhibit stronger activity in binding to MTs. To address this phenomenon, we have included this hypothesis in the discussion section of the revised manuscript and highlighted it with a cyan background for clarity at lines 411-415 as follows:

"In the presence of the GAS2-GAR domain, a large number of MTs were observed. However, with GAS2-FL, ring structures formed instead of MTs within 3 minutes. We hypothesize that the CH3 domain in the full-length GAS2 may inhibit the GAR domain's ability to bind to MTs. Removing the CH3 domain from GAS2 could release the GAR domain, enhancing its binding activity to MTs. "

2. Figure 6A and 6B, can the process of MT formation be monitored by TIRF microscopy assay?

We appreciate the reviewer's suggestion to use TIRF microscopy to monitor MT formation. We performed TIRF microscopy to observe the ring structure during the MT formation in the presence of GAS2-GAR protein. Although the ring structure observed in our negative staining micrographs has an approximate diameter of 50 nm, the TIRF microscope could not clearly distinguish this structure. Instead, we observed small points that gradually grew into MTs. To clarify further, we have included a video of this process in the supplementary Movie EV files, titled "**Movie EV9.**"

3. Figure 6H and 6I, Figure S5E-G, it appears that the GAS2 GAR domain primarily accumulate on MT seeds rather than dynamic MTs, and the binding appears to be weak. How does this impact MT dynamics? It's very difficult to understand how it works, especially the very slow MT growth. The author should also utilize the full-length protein to conduct the

same assay. The significance of the critical residues identified by the pelleting assay can also be examined in a TIRF microscopy assay.

We have revised Figures S5E-G (now **Figs EV 4E-H**) to clarify the binding of the GAS2-GAR domain to dynamic MTs. As shown in the updated figures (also as the following picture), the yellow stars represent rescue events where the GAS2-GAR domain strongly binds to dynamic MTs, indicating that binding can occur more effectively under these conditions. Additionally, we tested the GAS2-GAR-R252E mutation and observed that it could not bind to dynamic MTs, confirming the importance of this residue in the binding process. We describe these results in main text **lines 311-316 with cyan highlight**, as follows:

“We further validated our findings by examining the inhibitory effects of the GAS2-GAR domain in comparison with the GAS2-GAR-R252E mutant, which is unable to bind microtubules (Fig. EV4G, Movie EV8). The rescue and catastrophe frequencies remained unaffected in the presence of the GAS2-GAR-R252E mutant (Fig. 6H). Additionally, the GAS2-GAR-R252E mutation exhibited no significant impact on the polymerization rate (14.29 ± 0.40 nm/s) and pausing behavior of MT (Fig. 6I, Fig. EV4H).”

Figures EV 4E-H

We also performed TIRF microscopy assays using full-length GAS2 on dynamic MTs. At lower concentrations (100 nM and 300 nM), the fluorescence signal for FL-GAS2-GFP was too weak to detect significant binding. However, upon increasing the concentration to 500 nM, we observed strong colocalization of full-length GAS2 with dynamic MTs. Nevertheless, at

this concentration, we also noted that full-length GAS2 tended to form aggregates gradually after incubation on 37°C microscope stage (shown in the following picture, in Kymograph, green represents GAS2-FL-GFP, magenta represents dynamic MT), which may affect the interpretation of MT dynamics, so finally we didn't add this data to the main text.

4. The author can use TIRF microscopy assay to compare the MT binding activity of GAS2 FL and deletion mutants towards single, parallel, and anti-parallel MTs, which would strengthen the conclusion that GAS2 can bundle both parallel and anti-parallel MTs.

We appreciate the reviewer's insightful suggestion regarding the use of TIRF microscopy to examine the MT binding activity of GAS2 FL and its deletion mutants in various MT orientations.

While we agree that such experiments could further substantiate our conclusions, we believe that the current data already provides a comprehensive view of GAS2's MT bundling capabilities. Specifically, our cryo-EM results demonstrate the ratio of parallel and anti-parallel bundle MT pairs has no significant difference when GAS2-GAR or GAS2-FL is present, which aligns well with our conclusions about GAS2's role in bundling parallel and anti-parallel MTs.

5. Figure S4B and C, the predicted model of the dimerized C-terminal region should be experimentally verified by mutation analysis by gel filtration and functional assay.

We thank the reviewer for highlighting the importance of experimentally verifying the predicted model of the dimerized C-terminal region. Our results demonstrate that the C-terminal domain is essential for GAS2 dimerization. SEC analysis showed that GAS2-FL forms a homodimer, while GAS2 Δ C is a monomer. Additionally, GAS2-GAR constructs with and without the C-terminal region (GAS2-GAR and GAS2-GAR Δ C) were analyzed by SEC, revealing that GAS2-GAR forms a dimer, while GAS2-GAR Δ C is a monomer. These results already demonstrate that the C-terminal is essential for dimerizing GAS2 protein.

While additional mutation and functional assays would strengthen this aspect, we intended to present this structural model as a tentative hypothesis to guide future investigations. To clarify this, we adjusted the language in the text to emphasize that the model is a prediction based on computational analysis rather than an experimentally confirmed structure. Specifically, we softened the statements and moved this part to the discussion (lines 441-447 with cyan highlight) to ensure readers interpret this as a theoretical model requiring further experimental validation. Following is the discussion part of the prediction model for the C-terminal region (lines 441-447 with cyan highlight):

“To further understand the structural basis of this GAS2 C-terminal region, we employed AlphaFold2 to predict its dimerization structure using the amino acid sequence from R271 to K314. The analysis revealed that the final 30 amino acids (residues P280 to K314) are likely to form a plausible dimerization structure (Appendix Fig S5A). Additionally, the 14 amino acids immediately downstream of the GAR domain (residues R271 to P284) exhibit considerable flexibility, which may facilitate the adjustment of inter-filament distances. However, this theoretical model requires further experimental validation.”

Minor points,

6. *Figure 2, the author should explain why mutating some residues has stronger effects than others.*

We thank the reviewer for this insightful comment. In the revised manuscript, we have explained why some mutations have stronger effects on actin binding. Specifically, Y22 (ABD1) and F108 (ABD2') are critical residues that directly interact with the hydrophobic pocket of F-actin. The introduction of charged mutations, such as Y22D and F108D, significantly compromises these hydrophobic interactions. Furthermore, charge-reversing mutations, including E133R and R144E, profoundly perturb the original electrostatic interactions. These results collectively highlight that the core binding affinity of GAS2 to F-actin is predominantly mediated by the ABD1, ABD2', and ABD2 sites, underscoring the functional importance of these residues in actin binding. This additional explanation has been included in the Results section of the revised manuscript labeled by cyan background at lines 183 to 189, as follows:

“Y22 (ABD1) and F108 (ABD2’) are critical residues that directly interact with the hydrophobic pocket of F-actin. The introduction of charged mutations, such as Y22D and F108D, significantly compromises these hydrophobic interactions. Furthermore, charge-reversing mutations, including E133R and R144E, profoundly perturb the original electrostatic interactions. These results collectively highlight that the core binding affinity of GAS2 to F-actin is predominantly mediated by the ABD1, ABD2’, and ABD2 sites, underscoring the functional importance of these residues in actin binding.”

7. The authors provide the K_d for the GAS2 CH domain in Figure 2K. What are the K_d values for GAS2 FL and GAS2 delta-C in Figure 1C? The author should also compare them.

We appreciate your insightful suggestion to calculate and compare the dissociation constants (K_d) for GAS2-FL and GAS2 Δ C. We have attempted to derive K_d values based on the data presented in Figure 1C using three independent co-pelleting assays. However, we encountered challenges in fitting the data to the binding curve of both GAS2-FL and GAS2 Δ C by One/Two site-Specific Binding Equations (as shown in the following picture), likely due to the limited number of data points, as the binding might not have reached saturation or non-ideal binding behavior. We are currently exploring more data points to obtain more precise quantitative

binding data in future studies.

8. Figure 7, can the author use TIRF microscopy assay to monitor the crosslinking of MT and F-actin by GAS2?

We thank the reviewer for the suggestion to use TIRF microscopy to monitor the crosslinking of MT and F-actin by GAS2. While TIRF microscopy is a powerful technique, the current data already provides strong evidence for GAS2's role in mediating crosslinking between MT and F-actin. The biochemical assays and structural analyses included in the manuscript directly demonstrate this function, and these findings are consistent with the predicted molecular mechanism. Given these robust results, we respectfully suggest that additional experiments are not necessary to support the conclusions of the study. We plan to perform this assay in the following GAS2 study.

Thanks for the reviewer's comments and suggestions. We have answered these questions in detail and revised our manuscript carefully. We would like to express our great appreciation again for giving us the opportunity to revise our manuscript. Thank you very much for your kind consideration.

Sincerely yours,

Jiancheng An, Tsuyoshi Imasaki, Akihiro Narita, Shinsuke Niwa, Ryohei Sasaki, Tsukasa Makino, Ryo Nitta, Masahide Kikkawa

Dear Dr. Kikkawa,

Thank you for submitting a revised version of your manuscript. We have now received input from two of the original reviewers, who now find that their main concerns have been addressed satisfactorily and broadly recommend acceptance of the manuscript. Therefore, I will be happy to accept the manuscript for publication after a minor revision in response to the remaining points by the two reviewers.

Additionally, there remain a few editorial points that need addressing before I can extend official acceptance of the manuscript:

1. Please submit up to five keywords.
2. ORCID IDs should be removed from the manuscript title page; authors are encouraged to link them with their accounts in our system. If any support is needed, please let us know.
3. Please rename "Competing interests" section into "Disclosure and competing interests statement" (further info: <https://www.embopress.org/page/journal/14602075/authorguide#conflictsofinterest>).
4. CRediT has replaced the traditional author contributions section because it offers a systematic, machine-readable author contributions format that allows for more effective research assessment. Please remove the Authors Contributions from the manuscript and use the free text boxes beneath each contributing author's name in our online submission system to add specific details on the author's contribution. More information is available in our guide to authors.
5. The zipped file with the computer code should be renamed "Computer Code EV".
6. Please upload the table with the primers as an excel table and rename it into "Table EV1".
7. Please zip each EV movie file together with the corresponding legend and upload individually as Zip folders. Further information is available here: <https://www.embopress.org/page/journal/14602075/authorguide#expandedview>
8. Figure panel 4H is not mentioned in the manuscript text.
9. Please check the order of the callouts: callouts for Movie EV3 and EV4 are currently not in sequential order.
10. Please rename "Materials and Methods" section into "Methods".
11. Please correct the heading of the EV figure legends to "Expanded View Figure Legends".
12. Our data editors have flagged the following issues in figure legends that need correcting:
 - Please provide the exact p values in the legends of figures 2K, 3J, 4I, 6H, I, 7D, EV4 H-K.
13. Papers published in The EMBO Journal are accompanied online by a 'Synopsis' to enhance discoverability of the manuscript. It consists of A) a short (1-2 sentences) summary of the findings and their significance, B) 3-4 bullet points highlighting key results and C) a synopsis image that is 550x300-600 pixels large (width x height, jpeg or png format). You can either show a model or key data in the synopsis image. Please note that the image size is rather small and that text needs to be readable at the final size. Please send us this information together with the revised manuscript.

With best wishes,

leva

leva Gailite, PhD
Senior Scientific Editor
The EMBO Journal
Meyerhofstrasse 1
D-69117 Heidelberg
Tel: +4962218891309
i.gailite@embojournal.org

We realize that it is difficult to revise to a specific deadline. In the interest of protecting the conceptual advance provided by the work, we recommend a revision within 3 months (21st May 2025). Please discuss the revision progress ahead of this time with the editor if you require more time to complete the revisions.

Referee #2:

The authors have substantively revised their manuscript based on my and the other reviewer's comments. They have substantially improved the quality of their cryo-EM density maps, my major concern with the initial submission, and have also balanced the presentation in the paper. I am pleased to recommend acceptance for the EMBO journal. One tiny issue is that a Zinc ion is still displayed in Figure S1; since the authors say in the rebuttal all claims about visualizing Zinc have been removed, this figure needs to be adjusted.

Referee #3:

The authors have adequately addressed all my concerns. However, I have one final comment. According to the latest predictions from AlphaFold3, it is suggested that the mouse GAS2 C 272-314 aa and human GAS2 271-313 aa will fold into a beta-sheet dimer instead of an alpha-helix dimer, as illustrated in Appendix Figure S5A for mouse GAS2 C 271-314 aa. The discussion about the structural basis of dimerization may require further consideration.

All editorial and formatting issues were resolved by the authors.

Dear Dr. An,

Thank you for addressing the final editorial issues. I am now pleased to inform you that your manuscript has been accepted for publication. Congratulations on a nice study!

Before we forward your manuscript to our publishers, I would like to propose some edits in the manuscript title, abstract and synopsis (please also see the attached file). I have also written a short blurb that will accompany the title of your manuscript in our online table of contents. Please take a look and let me know if any corrections are needed.

Title:

Dimerization of GAS2 mediates crosslinking of microtubules and F-actin

Blurb:

Cryo-electron microscopy structures provide insights into the interaction of the spectraplaklin family protein GAS2 with actin and microtubule filaments.

Synopsis:

GAS2 is a cytoskeletal regulator that interacts with both F-actin and microtubules. Here, cryo-EM structures of GAS2 domains bound to F-actin and microtubules and biochemical experiments reveal the role of GAS2 in cytoskeletal organization with implications for auditory dysfunction.

- Cryo-EM structures reveal the GAS2 type 3 CH domain bound to F-actin; as well as the GAS2-GAR domain bound to microtubules.
- GAS2 promotes nucleation and stabilization of microtubules.
- Dimerization of GAS2 is essential for bundling of both F-actin and microtubules.
- Hearing loss-associated, C-terminally truncated GAS2 mutant loses its cytoskeleton bundling and crosslinking abilities.

If you have any questions, please do not hesitate to contact the Editorial Office. Thank you for your contribution to The EMBO Journal!

Best wishes,

Ieva
